# Identification of epilepsy-associated neuronal subtypes and gene expression underlying epileptogenesis

Ulrich Pfisterer[1,9], Viktor Petukhov[1,2,9], Samuel Demharter[1,9], Johanna Meichsner [3,9], Jonatan J. Thompson[4], Mykhailo Y. Batiuk [1], Andrea Asenjo Martinez[1], Navneet A. Vasistha [1], Ashish Thakur[1], Jens Mikkelsen[5], Istvan Adorjan[6], Lars H. Pinborg[7,8], Tune H. Pers [4], Jakob von Engelhardt[3], Peter V. Kharchenko [2] & Konstantin Khodosevich [1✉]

Epilepsy is one of the most common neurological disorders, yet its pathophysiology is poorly understood due to the high complexity of affected neuronal circuits. To identify dysfunctional neuronal subtypes underlying seizure activity in the human brain, we have performed single-nucleus transcriptomics analysis of >110,000 neuronal transcriptomes derived from temporal cortex samples of multiple temporal lobe epilepsy and non-epileptic subjects. We found that the largest transcriptomic changes occur in distinct neuronal subtypes from several families of principal neurons (L5-6_Fezf2 and L2-3_Cux2) and GABAergic interneurons (Sst and Pvalb), whereas other subtypes in the same families were less affected. Furthermore, the subtypes with the largest epilepsy-related transcriptomic changes may belong to the same circuit, since we observed coordinated transcriptomic shifts across these subtypes. Glutamate signaling exhibited one of the strongest dysregulations in epilepsy, highlighted by layer-wise transcriptional changes in multiple glutamate receptor genes and strong upregulation of genes coding for AMPA receptor auxiliary subunits. Overall, our data reveal a neuronal subtype-specific molecular phenotype of epilepsy.

[1] Biotech Research and Innovation Centre (BRIC), Faculty of Health and Medical Sciences, University of Copenhagen, 2200 Copenhagen, Denmark. [2] Department of Biomedical Informatics, Harvard Medical School, Boston, MA 02115, USA. [3] Institute of Pathophysiology, University Medical Center of the Johannes Gutenberg University Mainz, Mainz, Germany. [4] Novo Nordisk Foundation Center for Basic Metabolic Research, Faculty of Health and Medical Sciences, University of Copenhagen, 2200 Copenhagen, Denmark. [5] Department of Neurology and Neurobiology Research Unit, Copenhagen University Hospital, RigshospitaletCopenhagen, Denmark. [6] Department of Anatomy, Histology and Embryology, Semmelweis University, Budapest, Hungary. [7] Neurobiology Research Unit, Copenhagen University Hospital, Rigshospitalet 2200 Copenhagen, Denmark. [8] Epilepsy Clinic, Department of Neurology, Copenhagen University Hospital, Rigshospitalet 2200 Copenhagen, Denmark. [9] These authors contributed equally: Ulrich Pfisterer, Viktor Petukhov, Samuel Demharter, Johanna Meichsner. ✉email: konstantin.khodosevich@bric.ku.dk

Epilepsy is a neurological disorder that is characterized by spontaneous and reoccurring seizures that are mainly generated in the areas of the hippocampus or cerebral cortex[1,2]. Epilepsy remains the most common serious chronic disorder of the brain with more than 68 million people affected worldwide[3]. Active epilepsy is a devastating disorder that requires continued care, thus disrupting everyday aspects of life and imposing a physical, psychological, and social burden on patients and families[4].

The pathophysiology of epilepsy remains poorly understood. While there are some studies in animal models showing a contribution of certain neuronal subtypes to seizure generation and propagation[5], the corresponding data from human epilepsy patients are scarce. This can be explained by the complexity of neuronal networks involved in epileptogenesis. Recent studies suggest the presence of >60 neuronal subtypes in a single functional cortical area both in rodents and in human[6,7], and the same might be true for each area of the hippocampus, based on the number of GABAergic interneurons in the CA1 region in mice[8]. Importantly, not all neuronal subtypes will be similarly affected in epilepsy, and there is an indication from the literature that some subtypes are affected or contribute more to epileptogenesis than others. For instance, excitatory neurons of the CA1 region of the hippocampus are more affected than those from other hippocampal regions[9,10]. In addition, there is a decrease in the number of particular subpopulations of somatostatin (SST)- and neuropeptide Y (NPY)-positive GABAergic interneurons in the hippocampi of patients with temporal lobe epilepsy (TLE)[10]. In the cortex, selective impairment in gene expression of parvalbumin (PV)-positive GABAergic interneurons has been shown in epileptic tissue from focal cortical dysplasia type I/III, but not type II[11]. In a mouse model of seizure activation, different populations of GABAergic interneurons contribute to distinct stages of epileptogenesis[5]. These and other data[5,12] clearly show that depending on the disease phenotype, different neuronal assemblies and subtypes of neurons might be affected in epilepsy.

So far, gene-expression changes in epileptic brains have been studied in resected pieces of brain tissue to assess averaged changes across all types of neurons and glia as well as nonneural cells (blood vessels, ependymal cells, etc). Although such studies provided some important information about large-scale changes in gene expression, only relatively minor transcriptomic changes have been identified, even when comparing to highly sclerotic tissue[13–18]. This can be readily explained by the limitations of bulk-sequencing, where the averaging of gene expression in the tissue samples across all cell types leads to a diluted signal due to loss of information about specific cell types. Thus, to investigate how individual subtypes of neurons are affected in epilepsy, we study TLE by single-nucleus RNA sequencing (snRNA-Seq)[19–22] to process tissue from nonepileptic and epileptic human temporal cortex of multiple subjects using the 10X Chromium[23] and Smart-seq2[24] platforms. Our data reveal a differential effect of epilepsy on the neuronal transcriptome—while many subtypes exhibit mild gene expression changes, several specific subtypes of principal neurons and GABAergic interneurons are substantially affected. Strikingly, the most affected subtypes can be grouped based on commonality of epilepsy-related transcriptional changes, indicating that they belong to a circuit that might underlie epileptogenesis. In particular, multiple glutamate-signaling genes exhibit layer-wise dysregulation, suggesting different domains of the epilepsy-related neuronal network. Genes coding for AMPA receptor auxiliary subunits might be the strongest contributors to epileptogenesis owing to their vast layer-wise upregulation.

## Results

**Multipatient snRNA-Seq dataset for epileptic cortex.** To identify how distinct neuronal subtypes are affected and contribute to epilepsy, we analyzed epileptic cortical samples from patients with TLE using droplet-based 10X Genomics chemistry (Fig. 1a). We studied the temporal cortex since the focus of epilepsy in the hippocampal tissue in TLE patients usually shows severe degeneration, which hinders comparative transcriptomic analysis. Importantly, we chose only those patients that showed signs of abnormality in the temporal cortex identified by magnetic resonance imaging (MRI) indicating epilepsy-associated pathology (see "Methods" and Source Data Table 1). As controls, we used samples from the temporal cortices of subjects that did not have any neurological disorders. We microdissected 21 samples from epileptic and nonepileptic cortices that included all cortical layers, and sorted neuronal nuclei based on the expression of the neuronal marker NeuN (Supplementary Fig. 1a). Two control samples were excluded based on the low quality of the sequencing data: one sample did not show meaningful clustering, and in the other, a majority of cells expressed a high fraction (>20%) of mitochondrial transcripts. In total, we sequenced 117,221 nuclei, 101,982 of which passed quality control (see "Methods") and were recognized as neuronal subtypes. We also performed snRNA-Seq for the NeuN-negative fraction of four samples (two epileptic patients and two nonepileptic individuals) (Source Data Table 2). We confirmed that the vast majority of NeuN-negative nuclei came from glial and other nonneuronal populations (Supplementary Fig. 1b–e).

For further analysis, we utilized nine epileptic and ten nonepileptic cortices, with an average of 2304 genes detected per nucleus (Supplementary Fig. 2a, b and Source Data Table 2). We used Conos[25] to integrate all datasets and annotated them using established layer-specific markers for principal neurons and subtype-specific markers for GABAergic interneurons (Source Data Tables 3 and 4) (Fig. 1b and Supplementary Fig. 2c). One epilepsy (E5) and two nonepilepsy (C3 and C5) samples showed bias in integration (Supplementary Figs. 3 and 4) and the absence of a large number of subtypes (Supplementary Fig. 2d–g), and thus were excluded from the per-subtype analysis. The remaining samples were well integrated and showed a lack of experiment-batch effects (Supplementary Figs. 3 and 4). Age and sex had a low impact on the transcriptional identity of annotated neuronal subtypes (Supplementary Fig. 2h, i), although to rigorously study the effect of both of these parameters, a significantly larger sample size is required.

Initial analysis of single-nucleus transcriptomes by Conos revealed the known major transcriptomic subtypes of principal neurons and GABAergic interneurons in the temporal cortex (Fig. 1c). We carried out a hierarchical annotation strategy (Supplementary Fig. 5a, b) by separating all neurons into principal neurons and GABAergic interneurons, followed by separation of the major classes of principal cells using family-layer-specific marker genes (CUX2, RORB, THEMIS, and FEZF2). The interneurons were classified based on expression of cardinal markers genes PVALB, SST, VIP, and ID2 (Fig. 1c). In cases where several subtypes within a family could be grouped based on additional common markers, these were given a common subfamily name. Using this strategy, Fezf2 principal neurons could be further subgrouped into two subfamilies, Fezf2_Lrrk1 and Fezf2_Tle4 (Supplementary Fig. 5c), and Id2 interneurons could be further subdivided into two subfamilies, LAMP5-positive and LAMP5-negative (Supplementary Fig. 5d). Finally, families/subfamilies of principal neurons and GABAergic interneurons were classified into specific subtypes (Fig. 1b, d, e and Source Data Table 3). Clusters of principal neurons were validated using previously described layer-specific markers from healthy human cortex[7,26] that included LAMP5 (L2), PRSS12 (L2–L3), CUX2 (L2–4), RORB (L3–5), GRIN3A (L5), PCP4 (L5), HTR2C (L5), TLE4 (L5–6), GRIK3 (L5–6), OPRK1 (L5–6), and NR4A2 (L5–6)

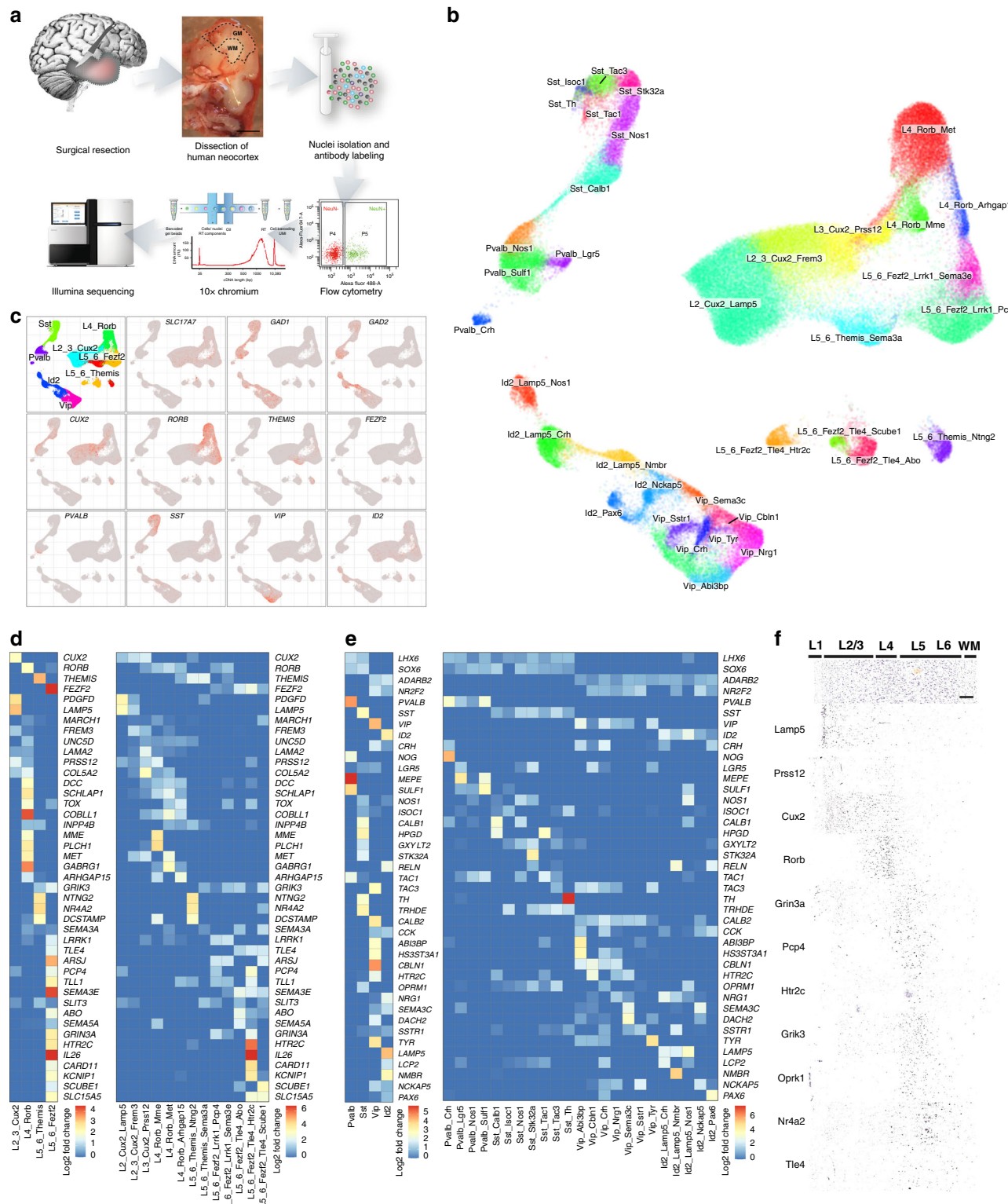

**Fig. 1 Multipatient single-nucleus transcriptomic dataset of the epileptic and nonepileptic temporal cortex. a** Schematic representation of the experimental outline for droplet-based single-nucleus RNA sequencing using 10× Chromium on FANS-isolated neuronal nuclei. Each sample was processed separately by FANS and 10× Chromium cDNA library preparation. **b** UMAP representation of neuronal nuclei isolated from multiple epileptic and nonepileptic cortices, and cell-type annotations for principal neurons and GABAergic interneurons. The colors represent subtypes with the labels showing subtype names. **c** General and family-specific marker expression for principal cells and GABAergic interneurons with the colors proportional to log-normalized expression values. **d**, **e** Family- and subtype-specific markers for principal cells (**d**) and GABAergic interneurons (**e**). Columns and rows represent subtypes and marker genes, respectively. The color shows the log2-fold change of this marker in a given subtype relative to the average expression in the other subtypes. **f** Confirmation of the layer-specific expression of cardinal markers for principal neurons in the healthy temporal cortex by in situ hybridization (taken from Allen Brain Atlas)[7,22]. Scale bar: 400 μm.

(Fig. 1f). Overall, we annotated 13 principal neuron and 23 GABAergic interneuron transcriptomic subtypes (note that the layer position for principle neurons is predictive). Our annotation of the healthy human temporal cortex matched well with a study that used the Smart-seq method and proposed ~70 neuronal subtypes in this cortical area[7]. In many cases, one of our subtypes matched to more than one subtype in the previous dataset (Supplementary Fig. 6a–d), which can be explained by a higher resolution of Smart-seq relative to the 10× Genomics method. In addition, to account for human heterogeneity, each annotated transcriptomic subtype in our data has at least five cells in at least 90% of the samples (Supplementary Fig. 2d, e). The only exception is the rare Sst_Th subtype, which had very few cells in general, thereby limiting the power of our conclusions about this subtype.

To confirm that none of the annotated transcriptomic subtypes were selectively lost in the NeuN-negative fraction, we analyzed our previously sequenced NeuN-negative fractions and showed that only a very small proportion within this population displayed a neuronal identity (Supplementary Fig. 1c–e).

To confirm that the differences in gene expression and subtype composition between healthy and epileptic cortices did not depend on the snRNA-Seq method, we profiled the transcriptomes of 1114 single neuronal nuclei for one of the epileptic patients using a modified version of Smart-Seq2 (see "Methods") (Supplementary Figs. 1f and 7a, and Source Data Table 2). On average, 7000 genes/nucleus were detected (Supplementary Fig. 7b, c), and no substantial experiment-batch effects were observed (Supplementary Fig. 7d). We were able to identify all subtypes that were previously identified and annotated using droplet-based 10× Genomics method (Supplementary Fig. 7f, g), and the relative distribution of nuclei assigned to discrete subtypes showed a high similarity between the two methods (Supplementary Fig. 7h), confirming that subtype identification was comparable between the Smart-seq2 and 10× datasets.

**Disease-related neuronal subtypes in the epileptic cortex.** In order to identify those subtypes of neurons that were either affected by or contributed to epilepsy, we compared our snRNA-Seq data from the epileptic temporal cortices with nonepileptic cortices. Both datasets were processed simultaneously using the same protocol for cDNA library preparation, thereby limiting batch effects. Although the joint healthy and epileptic dataset contains both autopsy and biopsy samples of the temporal cortex, such an integration has been done successfully before[7]. Only a small number of genes were attributed to either post-mortem (autopsy) or injury (biopsy) signatures of the temporal cortex[7] (Source Data Table 5). In our dataset, autopsy samples had a very low postmortem interval (Source Data Table 1), and the proportions of recovered nuclei were similar across transcriptomic subtypes between biopsies and autopsies (Supplementary Fig. 8a).

Overall, the integration of epileptic and nonepileptic temporal cortices revealed generally good agreement between the subtypes identified in the two datasets. However, in several locations, for instance, upper cortical layers L2_Cux2_Lamp5 and L2–3_Cux2_Frem3, there was a visible transcriptomic shift between neurons derived from epileptic and nonepileptic cortices (Fig. 2a). In addition, although the number of neurons for each subtype was approximately similar between epilepsy and nonepilepsy samples, we observed a notable decrease in the number of identified nuclei for several subtypes (Fig. 2b). Thus, the numbers of L2/3 subtypes were reduced, and the reduction was even more pronounced when we normalized for the total number of sequenced neurons for each condition (Fig. 2c). For interneurons,

the largest decrease in neuronal number was observed for the Pvalb_Sulf1 subtype (Fig. 2c).

In order to find disease-affected transcriptomic subtypes in the epileptic tissue, we performed a gene-expression correlation analysis between neuronal subtypes of normal and epileptic datasets. We used a gene-expression-similarity score based on the Pearson correlation of expression within and between conditions (see "Methods"), where a low similarity value highlights a large difference between epileptic and nonepileptic neuronal subtypes (Fig. 2d). Importantly, this score does not depend on the cluster size, gene, and read number per cluster/subtype (Supplementary Fig. 8b–d). Based on this score, some subtypes of the epileptic cortex had a relatively low similarity with their counterparts in the control samples (Fig. 2d), which might indicate an epilepsy-specific effect on their transcriptome. For principal neurons, the largest transcriptomic differences between epileptic and nonepileptic cortices were noted for the L5–6_Fezf2_Tle4_Abo, L5–6_Themis_Ntng2, L2_Cux2_-Lamp5, and L2–3_Cux2_Frem3 subtypes (Fig. 2d). For GABAergic interneurons, the largest transcriptomics alterations in the epileptic temporal cortex were found in Vip_Cbln1, Sst_Tac1, Pvalb_Sulf1, Pvalb_Nos1, and Id2_Lamp5_Nos1. Notably, every cardinal class of GABAergic interneurons was sharply separated into more and less-affected parts, which highlights possible selective vulnerability of individual interneuronal subtypes within a class to epilepsy. For instance, for Id2 neurons, the *LAMP5*-positive subfamily showed a larger epilepsy-related effect on their transcriptome than the *LAMP5*-negative subfamily (Fig. 2d), whereas for Vip neurons, Vip _Cbln1, Nrg1, Sema3c, and Tyr had a higher divergence of epileptic versus nonepileptic transcriptomes versus Vip_Abi3bp, Crh, and Sstr1, respectively.

To validate that the observed differences between neuronal subtypes in nonepileptic and epileptic cortices were disease-related, we went on to test if the transcriptomic changes were enriched in genes associated with epilepsy. We calculated the differentially expressed (DE) genes between all TLE and non-TLE cortical samples for each subtype using an adapted DESeq2 method in Conos (Source Data Table 6), and also visualized P value and fold change for each DE gene in an unbiased way for each neuronal subtype by volcano plots[27] (Supplementary Figs. 9 and 10, for principal neurons and GABAergic interneurons, respectively). Next, we analyzed the DE genes for enrichment in epilepsy-associated genes using two gene lists, one curated based on genetics studies in human patients and mouse models and the other derived from the largest epilepsy genome-wide association study (GWAS) to date[28] (Source Data Tables 7 and 8, respectively). This enrichment analysis confirmed an overall prevalence of epilepsy-associated DE genes in neuronal subtypes with larger transcriptomic differences identified by our gene expression correlation analysis (Fig. 2e, f and Source Data Table 9).

**Shared and subtype-specific epilepsy-associated pathways.** Based on previous genetics and functional studies, we expected epilepsy to coincide with both pan-neuronal and subtype-specific changes. Furthermore, there are existing data that highlight differential vulnerability/contribution of neuronal subtypes to epilepsy[5,10,11]. Thus, we investigated both epilepsy effects that were shared across neuronal subtypes and those that were specific for individual subtypes. To identify gene categories that changed their transcriptomic pattern in neuronal subtypes of the epileptic cortex, we calculated the enrichment of DE genes for Gene Ontology (GO) terms in each of the identified subtypes (Source Data Tables 10 and 11). The GO terms that shared a large fraction of the enriched genes were grouped together under the name of the most significant GO term of the

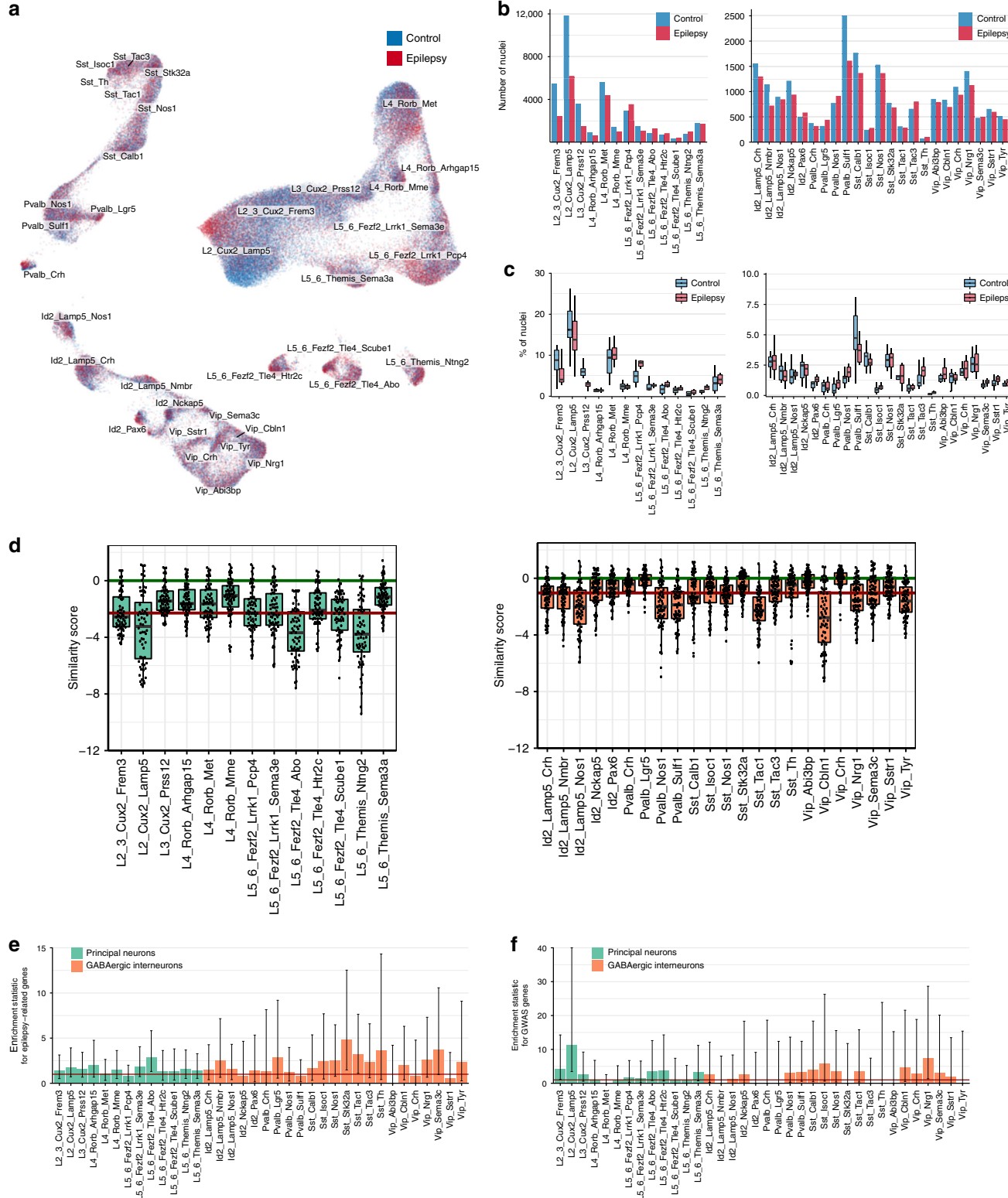

group in terms of enrichment (Source Data Table 10 and "Methods"). Interestingly, whereas several neuronal subtypes exhibited large transcriptomic changes in epilepsy (>100 enriched GO terms), a large group of neuronal subtypes had only few enriched GO terms, in particular for biological process (BP) terms that should be related to biological function of a pathway (Fig. 3a). The majority of these less-affected subtypes also had the highest gene expression correlation between epileptic and

nonepileptic cells (Fig. 2d), thus confirming that subtypes with high transcriptomic correlation between epilepsy and none-pilepsy also had fewer signaling pathways changed by epilepsy. Although there was some correlation between the number of GO terms to the number of DE genes and number of DE genes to the number of sequenced cells, no correlation was observed between the number of GO terms and the number of sampled cells in a subtype (Supplementary Fig. 11a–c).

**Fig. 2 Integration of epileptic and nonepileptic datasets and identification of disease-related neuronal subtypes. a** UMAP embedding with the integration of epileptic and nonepileptic datasets using Conos, colored by condition. **b** The total number of nuclei identified per subtype and condition. **c** Percentage of nuclei per subtype, showing compositional change across conditions. A notable decrease in L2/3 subtypes as well as Pvalb_Sulf1 was observed for epilepsy. **d** Similarity score, based on gene-expression correlation between neuronal subtypes in the epileptic and nonepileptic cortex that reveals disease-related subtype-specific transcriptomic changes in the epileptic tissue. A lower similarity score indicates larger differences between conditions. The red line indicates the median similarity score across all subtypes. The green line represents the 0 level that corresponds to "no difference observed". **e**, **f** Analysis showing overrepresentation of differentially expressed (DE) genes between epileptic and nonepileptic datasets in genes identified in genetic studies in human patients and mouse models (**e**), and epilepsy genes identified in the largest epilepsy GWAS study to date (**f**)[23]. The odds ratio of the Fisher's test is shown on the y scale with the bar height corresponding to the conditional maximal likelihood estimate and whiskers showing 95% confidence intervals. The red horizontal line shows an odds ratio equal to 1, which corresponds to "no difference observed".

Clustering the GO terms by their level of enrichment in each of the subtypes revealed sets of subtypes showing similar enrichment patterns for specific terms. This clustering structure suggests underlying common transcriptomic shifts in groups of neurons, indicating that they may be part of the same circuit/assemblies (Fig. 3b and Supplementary Figs. 12–15). The most prominent shifts included GO terms involved in neural circuit reorganization and neurotransmission, which were grouped in the "Pink Cluster": regulation of membrane potential, modulation of chemical synaptic transmission, synapse assembly, synapse organization, axon development, regulation of neuron projection development, dendrite development, and few others (Fig. 3b and Supplementary Fig. 12). These shifts were observed in principal neurons from all layers and a few specific interneuron subtypes: Id2_Lamp5 subfamily, Sst_Tac1, Sst_Tac3, and Vip_Cbln1.

To gain further insight into transcriptomic changes in the Pink Cluster, we again clustered neuronal subtypes, now based on the similarity of DE genes from this cluster. This analysis showed that synapse assembly and synapse organization DE genes were in general similar for principal neurons, but differed for interneurons (13, 14 in Fig. 3c), which was also confirmed when we clustered the individual DE genes for synapse assembly and synapse organization showing several DE gene sets that were specific either for principle neurons or interneurons (Supplementary Fig. 13a, b). Interestingly, principal-neuron synapse organization DE genes clustered with dendritic spine development DE genes (13, 14, and 4 in Fig. 3c and Supplementary Fig. 13a–c). Principal neurons again clustered separately for dendrite development DE genes (3 in Fig. 3c and Supplementary Fig. 14a), and similar GO terms for interneurons were rather scattered. However, axon development DE genes exhibited co-clustering between principal neurons and interneurons (1 in Fig. 3c and Supplementary Fig. 14b). Finally, strong co-clustering for genes involved in regulation of membrane potential was observed for principal neurons and all major classes of interneurons (9,11 in Fig. 3c and Supplementary Fig. 15a, b).

Further reclustering of DE genes for each GO term within the Pink Cluster might help in finding local circuits that have common epilepsy-related transcriptomic changes. Examples include L2–3_Cux2_Frem3 and L2_Cux2_Lamp5 upper-layer neurons together with L5–6_Themis_Sema3a and L5–6_Themis_Ntng2 lower-layer neurons (Supplementary Figs. 13a, b, 14a, b, and 15c), multiple subtypes of principal neurons with Sst interneurons (Supplementary Figs. 13b and 15a, b), and L2–3_Cux2_Frem3 and L2_Cux2_Lamp5 principle neurons with Vip_Cbln1 interneurons (Supplementary Fig. 15c). Finally, we plotted all neuronal subtypes based on similarity of the enriched GO terms, allowing us to identify several groups of subtypes, where each group might underlie a local circuit (Fig. 3d).

Reorganization of neuronal circuits was also highlighted by GO terms in other clusters. The Blue Cluster included multiple terms that were related to developmental processes, ion transport, and glutamate signaling (Fig. 3b and Supplementary Fig. 16a), which mainly affected L2_Cux2_Lamp5, L4_Rorb_Met, and multiple L5–6 subtypes of principal neurons, as well as Id2_Lamp5 and Sst_Tac1 and Sst_Tac3 interneurons. The Violet Cluster included mainly GO terms for protein transport to axons/ dendrites that were selectively affected in L4_Rorb principal neurons (Fig. 3b and Supplementary Fig. 16b). The Dark Green Cluster was specific for L4_Rorb_Met, L5–6_Fezf2_Tle4_Abo, and L5–6_Themis_Sema3a that co-clustered with Id2_Lamp5_Nmbr and included GO terms associated with cell adhesion, ion transport, and synaptic plasticity (Fig. 3b and Supplementary Fig. 16c). Finally, the Light Green Cluster highlighted the selective effect of epilepsy on upper-layer L2–3_Cux2_Lamp5 and L2–3_Cux2_Frem3 principal neurons that consist of dysregulated neuronal morphogenesis GO terms (Fig. 3b and Supplementary Fig. 16d).

Importantly, in line with the above GO-term analysis, disease ontology (DO) queries revealed multiple DO terms associated with epilepsy that were dysregulated in our snRNA-Seq dataset (Source Data Table 12). In particular, several Sst subtypes, Vip_Cbln1 and Id2_Lamp5 subtypes showed high enrichment for DO terms such as focal epilepsy (DOID:2234), epilepsy syndrome (DOID:1826), and temporal lobe epilepsy (DOID:3328) (Source Data Table 12). Furthermore, we found the same epilepsy-associated DO terms enriched in L3_Cux2_Prss12, L5–6_Fezf2_Lrrk1_Sema3e, and L5–6_Fezf2_Tle4_Abo subtypes of principal neurons. This shows that DO terms associated with epilepsy are enriched differentially in different neuronal subtypes, with epilepsy-related DO terms being more broadly enriched in GABAergic compared to glutamatergic cells and with Sst subtypes, Vip_Cbln1 and Id2_Lamp5, revealing particular vulnerability to epilepsy.

To explore differential neuron-type vulnerability in epilepsy and neuron-type-specific response to epilepsy, we searched for GO terms and DE genes that were unique for particular neuronal subtypes in the epileptic cortex. The majority of neuronal subtypes showed enrichment for specific GO terms in their DE genes (Supplementary Fig. 17 and Source Data Table 13). Moreover, few subtypes were particularly enriched for specific GO terms, including L5–6_Fezf2_Tle4_Abo (e.g., G protein-coupled gluta-mate receptor signaling pathway, GO:0007216; ephrin receptor signaling pathway, GO: 0048013), L2–3_Cux2_Lamp5 (e.g., lysosome organization, GO:0007040; insulin receptor signaling pathway, GO:0008286), Id2_Lamp5_Nmbr (e.g., nitric oxide-mediated signal transduction, GO: 0007263), and Sst_Tac1 (e.g., netrin-activated signaling pathway, GO:0038007) subtypes.

**Signaling pathways underlying seizure activity.** Although many differences in the transcriptome between epileptic and none-pileptic cortices might be consequences rather than the causes of the disease, some of the differences should contribute to a distinct feature of epilepsy—generation of epileptic seizures. This in turn can be caused by hyperexcitability of principal neurons and/or hypoinhibition of principal neurons from GABAergic

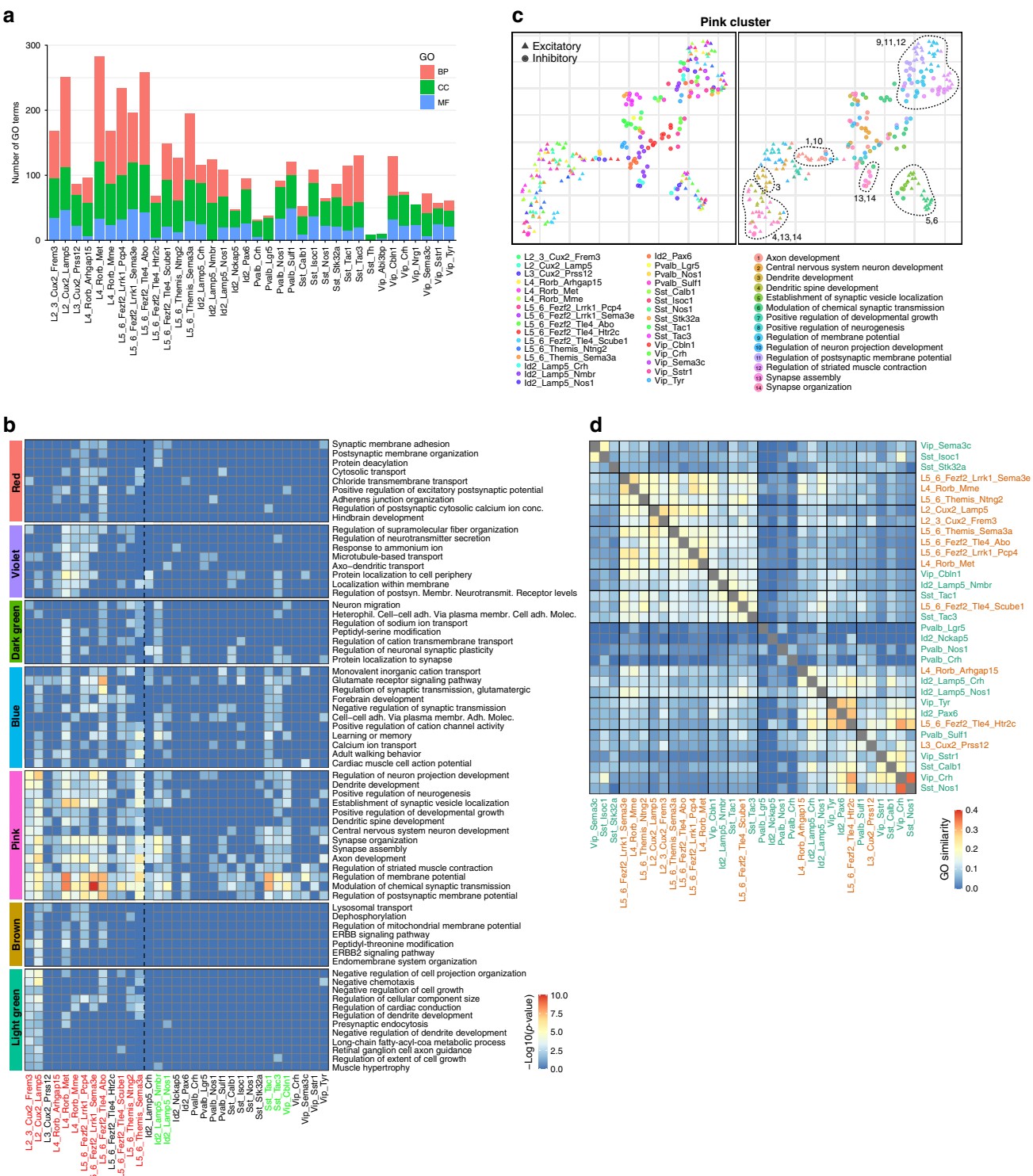

interneurons. In order to find candidate neuronal subtypes contributing to seizure activity, we searched for GO terms that could contribute to hyperexcitability of principal neurons or hypoinhibition by GABAergic interneurons. Glutamate receptors and in particular AMPA receptors were shown to be one of the major drivers of seizures[29,30] and their antagonists are antiepileptic[31,32]. Thus, we chose GO terms GO:0001508 "action potential" and GO:0007215 "glutamate receptor signaling pathway" as well as GO:0032281 "AMPA glutamate receptor complex" to search for subtypes enriched in hyperexcitability-related transcriptomic changes and plotted DE genes from these GO terms after filtering

for genes with lack of expression (Fig. 4a–d and Source Data Table 10). Although such an approach is based on previous findings, it should also allow us to identify novel genes that are dysregulated in epilepsy and might contribute to seizure activity. Indeed, a number of the genes with the largest changes in expression have not previously been reported for epilepsy. In particular, AMPA auxiliary subunit CKAMP44 (*SHISA9*), a member of the recently discovered family of AMPA receptor auxiliary subunits[33,34] that increases current amplitudes for AMPA receptors[35], had increased expression across almost all layers of principal neurons (Fig. 4c, d). Most pronounced

**Fig. 3 Identification of epilepsy-associated pathways and transcriptomic shifts across neuronal subtypes. a** GO-term enrichment analysis ordered by neuronal subtype reveals both subtypes with large transcriptomic changes (>100 GO terms) and subtypes with only few or no enriched GO terms in the epileptic dataset. The total number of GO terms that passed the 0.05 threshold for the adjusted *P* value of the overrepresentation test is shown on the *y* axis. Colors of the stacked barplot represent the top-level GO term: biological process (BP), cellular component (CC), or molecular function (MF). **b** The major groups of GO terms clustered by their level of enrichment per subtype reveal common transcriptomic shifts across neuronal subtypes in the epileptic brain. Rows correspond to GO terms, ordered according to hierarchical clustering. Columns correspond to cell types. Colors represent –log10 of adjusted *P* values of the overrepresentation test, trimmed with the upper boundary of 10. **c** For the blue cluster in (**b**), the plot shows a UMAP embedding of the GO terms per each subtype. Each point corresponds to a single square on the heatmap in (**b**). The distances between points are proportional to the Jaccard distance of the enriched genes between two given GO terms in certain subtypes. Thus, points, which are close to each other on the plot, are represented by similar sets of the enriched genes. Left—colored by subtype, right—colored by GO term. The numbers on the right panel indicate GO terms in a subcluster outlined by the dashed line. **d** Heatmap showing neuronal subtypes grouped based on Jaccard similarity of the enriched "Biological Pathway" GO terms. Rows and columns correspond to cell types, and the intersection represents the weighted Jaccard similarity between the two subtypes. Bold lines separate high-order clusters; neuronal subtypes labeled by the orange and green colors correspond to GABAergic interneurons and principal neurons, respectively. Such a clustering allows to identify groups of subtypes, where each group might correspond to a local circuit/network.

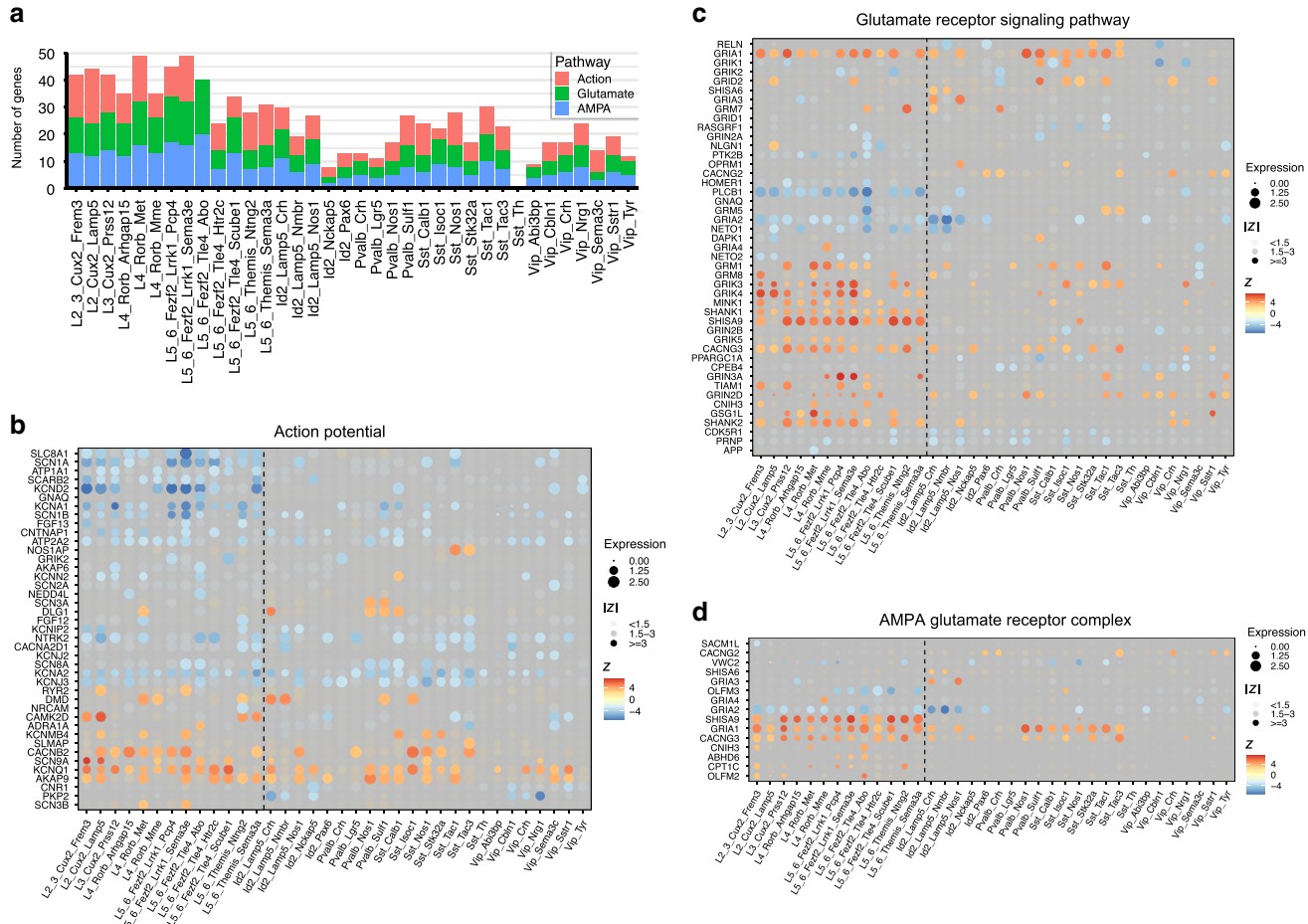

**Fig. 4 Identification of signaling pathways and genes in cortical neuronal subtypes that might underlie seizure activity. a** The total number of enriched DE genes for GO terms "action potential", "glutamate receptor signaling pathway", and "AMPA glutamate receptor complex" in each neuronal subtype. **b**–**d** Expression level of DE genes found in the GO terms "action potential", "glutamate receptor signaling pathway", and "AMPA glutamate receptor complex", respectively, ordered by neuronal subtype. The color of the points represents *Z* scores of differential expression between conditions, with the blue color showing downregulation and red colors indicating upregulation of gene expression in epilepsy. The size of the points corresponds to the average expression level of a gene in a given cluster. Points with low *Z* scores have higher transparency.

upregulation (>twofold change, *Z* score >7), occurred in L5–6_Fezf2 and L4_Rorb principal neurons. The importance of CKAMP44 upregulation is also highlighted by the fact that CKAMP44 is the main AMPA receptor auxiliary subunit across the whole temporal cortex (Fig. 4c, d). Interestingly, we observed a rather general effect of epilepsy on the upregulation of genes coding for AMPA receptor auxiliary subunits. Thus, AMPA auxiliary subunits CKAMP52 (*SHISA6*), TARP-γ2 (*CACNG2*), TARP-γ3 (*CACNG3*), cornichon 3 (*CNIH3*), and GSG1L (*GSG1L*) were upregulated in one or several principal neuron and interneuron subtypes (Fig. 4b). Of these, rather specific upregulation was observed for CKAMP52 in Id2_Lamp5 subtypes and cornichon 3 in L5–6_Fezf2_Tle4_Abo, L2–3_Cux2_Frem3, and L4_Rorb_Met, respectively.

There was complex dysregulation of expression for multiple glutamate receptor subunits and neuronal activity-related genes, and the majority of these changes were not reported for epilepsy before. Thus, while most of genes coding for glutamate receptor subunits increased their expression in epilepsy (*GRIA1*, *GRIA3*, *GRIA4*, *GRIK3*, *GRIK4*, *GRIK5*, *GRIN2B*, *GRIN3A*, *GRM1*, *GRM7*, and *GRM8*), few were downregulated (*GRIA2*, *GRIA3*, *GRIN2A*, *GRM5*, *GRIK1*, and *GRIK2*) (Fig. 4b, c). Furthermore, some subunits had rather specific up- or downregulation, e.g., *GRIA4* was upregulated specifically in L4_Rorb_Mme, whereas others were more widespread, e.g., *GRIA1*, *GRIK3*, and *GRIK4* (Fig. 4c, d). In particular, *GRIA1* was upregulated in the majority of principal-neuron subtypes as well as several Pvalb and Sst subtypes of interneurons. Furthermore, a number of genes regulating neuronal activity were downregulated across principal neurons, such as multiple voltage-gated sodium and potassium channels (*SCN* and *KCN* genes), *PLCB1* and *PTK2B* (Fig. 4b, c).

In spite of complex changes in glutamate signaling in epilepsy, it was clear that several subtypes accounted for the largest epileptic effect. In particular, L5–6_Fezf2_Tle4_Abo displayed downregulated *GRIA1*, *GRIA2*, *GRIN2A*, and *GRM5* subunits, and upregulated *SHISA9*, *CNIH3*, *GRIA1*, *GRIN3A*, *GRIK4*, *GRM1*, and *GRM7*, whereas both *GRIA1* and *GRIA4* were upregulated in L2–3_Cux2_Frem3 in addition to *SHISA9*, *CACNG3*, *CNIH3*, and *GRIK3* (Fig. 4c, d). For interneuron subtypes, the largest glutamate-signaling dysregulation was in epileptic Sst_Tac1 and _Tac3 subtypes that upregulated both TARP-γ2 and TARP-γ3 as well as *GRIN3A*, *GRM1*, *GRM5*, *GRIN2D*, and *GRIK3*.

Interestingly, we noted that the majority of genes involved in glutamate-mediated excitation showed layer-wise changes in expression in the epileptic brains (Fig. 5a). Such a structured pattern of dysregulation of glutamate excitation genes in epilepsy suggests the existence of several neuronal networks with discrete effects resulting from epilepsy, where a combination of these networks contributes to hyperexcitability. To confirm such a structured and layer-specific dysregulation of gene expression in the cortex of epileptic patients, we labeled mRNAs for several highly modulated genes by single-molecule fluorescent in situ hybridization (smFISH) method. As we reported above, a number of genes coding for AMPA auxiliary subunits were upregulated in a layer-wise manner in subtypes of principal neurons. Thus, we labeled CKAMP44 mRNA in a set of cortical sections from epileptic and nonepileptic samples, and by co-labeling with Rorb and DAPI, we identified the positions of L2/3, L4, and L5/6 (Fig. 5b). Importantly, we confirmed a dramatic upregulation of CKAMP44 expression across the layers (Fig. 5b), and in addition quantified upregulation in Rorb+ neurons in the L4–5 (Fig. 5c, d) (that represent all three subtypes of L4_Rorb neurons and two subtypes of L5–6_Fezf2_Lrrk1 family—see Rorb expression in Fig. 1d).

Another class of genes that exhibited dramatic layer-wise dysregulation in epileptic cortex were coding for glutamate receptor subunits (Figs. 4c and 5a). Thus, we selected *GRIA1* and *GRIN3A* as those of the most upregulated and validated their change in expression in epilepsy by smFISH. We partitioned cortical sections from nonepileptic and epileptic patients into layers based on DAPI (Fig. 6a), and performed quantitative analysis of *GRIA1* expression in L2–3 and L5–6 and *GRIN3A* expression in L5–6 (since in snRNA-Seq, the former showed upregulation in both upper and lower layers, whereas the latter in lower layers). Importantly, smFISH confirmed ~2- and ~3.5-fold increase in *GRIA1* expression in L2–3 (Fig. 6b, e) and L5–6 (Fig. 6c, f), respectively. Furthermore, there was approximately threefold upregulation of *GRIN3A* expression in L5–6 (Fig. 6d, g). Overall, smFISH validated layer-wise and subtype-specific changes in gene expression that were identified by snRNA-Seq.

For GO terms that might be related to hypoinhibition of principal neurons from GABAergic interneurons, we searched mainly for expression of *GAD1* and *GAD2*, key enzymes in GABA synthesis. There was an overall decrease in expression of both genes with some interneuron subtypes affected more than others, e.g., Pvalb family neurons for *GAD1* and Id2 for *GAD2* (Fig. 7a, b). Interestingly, we noted highly specific expression of cannabinoid receptor 1 (*CNR1*, CB1 for protein) in the cortex, which is involved in presynaptic inhibition of GABAergic interneurons[36]. To this end, *CNR1* was expressed in several Vip and Id2 non-Lamp5 subtypes, with very little expression in other subtypes of GABAergic interneurons and principal neurons (Fig. 7c and Source Data Table 6). The expression of *CNR1* was decreased in TLE patients for both Vip and Id2 non-Lamp5 subtypes. We confirmed the decrease in *CNR1* expression in Vip subtypes by smFISH (Fig. 7d, e). Thus, lower expression of *CNR1* might lead to increased activity of Vip interneurons and thus disinhibition (through inhibition of other inhibitory neurons) of principal neurons.

**Gene networks that are affected by epilepsy.** To get further insight into putative biological pathways transcriptionally associated with epilepsy, we identified sets of co-expressed genes (henceforth referred to as gene modules) within major cell types (Source Data Table 14). We performed robust weighted gene co-expression network analysis (rWGCNA)[37,38] within each of the level-2 cell types, and identified 12 gene modules robustly associated with epilepsy status (Supplementary Fig. 18a and Source Data Table 15), after discarding modules that overlapped with larger modules (Source Data Table 16), enriched for postmortem or injury-associated transcriptional biases (Source Data Table 5, taken from previous study[7]) or whose expression profile was better explained by sample than by epilepsy status. Six of the modules robustly associated with epilepsy were upregulated and six were downregulated in epileptic samples (Supplementary Fig. 18b).

To identify the cells in which the modules were most tightly linked, we measured the extent to which co-expression was preserved within each of the level-4 subtypes of the level-2 cluster, in which the module was detected. We found that several of the modules that were associated with epilepsy were particularly highly co-expressed in cell types implicated by our analyses above, notably the principal neurons L5–6_Fezf2_Tle4_Abo, L5–6_Fefz2_Lrrk1_Sema3e, and all three L2–3_Cux2 subtypes (e.g., *darkorchid4*, *mediumorchid*, *royalblue1*, *antiquewhite*, and *orange1* in Source Data Table 19).

We subsequently tested each prioritized module for its co-expression with the curated list of genes associated with epilepsy based on genetics studies in human patients and mouse models (Source Data Tables 7 and 18). Among the 12 gene modules tested, 7 were found to be enriched for genes from the curated epilepsy gene list ("Methods"; Supplementary Fig. 17a). Gene set enrichment analysis of these seven modules across the GO database ("Methods") highlighted, among others, gene sets related to neuron part, synapse, nervous system development, and plasma membrane part for the *olivedrab2* module; cyclic-nucleotide binding, cyclic-nucleotide phosphodiesterase activity, and plasma membrane for the *mistyrose3* module; synaptic membrane and presynaptic membrane for the antiquewhite1 module; neuron projection membrane and diacylglycerol kinase activity for the *bisque2* module; plasma membrane, glutamate receptor activity, and synapse for the azure4 module; ion channel activity and

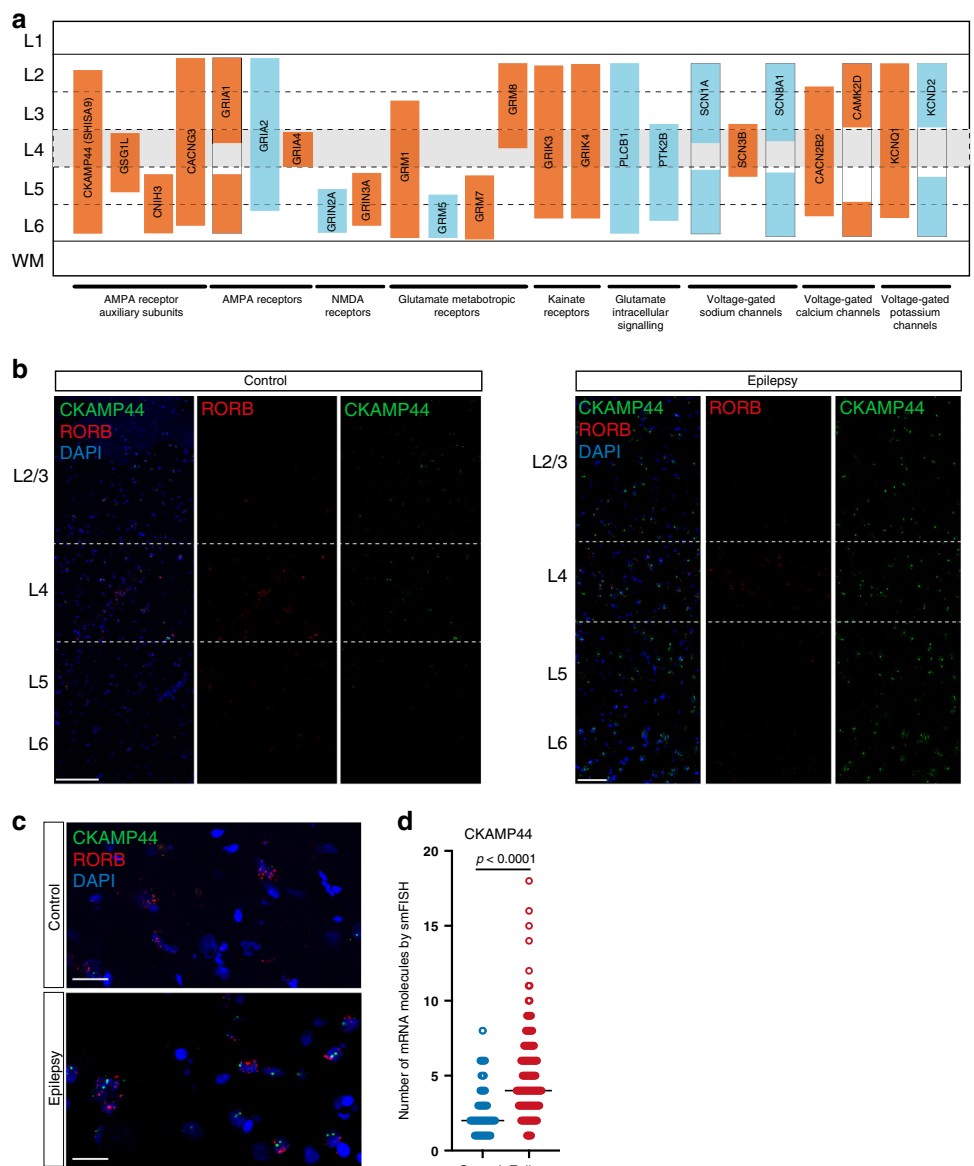

**Fig. 5 Complex layer-wise dysregulation of genes involved in glutamate-mediated excitation in the cortex of epileptic patients. a** Notable layer-wise upregulation or downregulation of genes involved in glutamate-mediated excitation. **b** Overview of smFISH that shows CKAMP44 (*SHISA9*) and RORB mRNAs in the temporal cortex of control (nonepileptic) and epileptic brains. The RORB probe was used to label layer L4, thereby allowing the identification of the cortical layer structure (together with DAPI staining). Note the higher number of CKAMP44 mRNA molecules across all layers in the epileptic brain. **c, d** Representative image and quantification of CKAMP44 mRNA molecules in control (nonepileptic) and epileptic temporal cortices demonstrate upregulation of CKAMP44 in RORB-positive neurons belonging to L4_Rorb and L5–6_Fezf2_Lrrk1 principal neurons (one-tailed Mann–Whitney test, 98 cells in two sections of two control brains and 215 cells in three sections of three epileptic brains).

inorganic ion transmembrane transport for the *mediumpurple3* module; neuronal action potential and voltage-gated sodium channel complex for the *royalblue* module (Source Data Table 19).

Several genes highlighted in the above cell-level analyses and involved in hyperexcitatory signaling and seizures were also members of one or more of the epilepsy-associated gene modules. Notably, with respect to AMPA receptor auxiliary subunits, the *darkorchid4* (L5–6_Fezf2_Tle4_Abo) and *chocolate1* (L4_Rorb_Schlap1_Mme) modules, both upregulated in epilepsy, contained the *SHISA9* gene (CKAMP44); the *darkorchid4* module also included the *CACNG3* gene (TARP-γ3), while the *GSG1L* gene was part of the *royalblue1* (L2_Cux2_Lamp5), *chocolate1* (L4_Rorb_Schlap1_Mme), and *olivedrab2* (L5–6_Themis_Ntng2) modules (Source Data Table 15).

Genes coding for glutamate receptor subunits were also widely represented within the epilepsy-associated modules. Notably, the *royalblue1* (L2_Cux2_Lamp5) and *darkorchid4* (L5–6_Fezf2_Tle4_Abo) modules, both similarly upregulated in epilepsy, shared nine genes including *GRIK4* and *GRIA1*. The *azure4* module (L4_Rorb_Schlap1_Met, upregulated in epilepsy) included the *GRIK3* and *GRM8* genes. Additionally, the *mediumblue* module (Pvalb subtypes, 173 genes), while not passing the test for associating more strongly with epilepsy than with any sample, included the genes *GRIK1*, *GRIK2*, *GRIK4*, *GRM1*, *GRM7*, and *GRIA1*, suggesting a potential involvement in glutamate receptor subunits. The voltage-gated sodium gene *PLCB1* was included in the *darkgoldenrod1* (L2–3_Cux2) and *olivedrab2* (L5–6_Themis_Ntng2) modules, both downregulated in epilepsy.

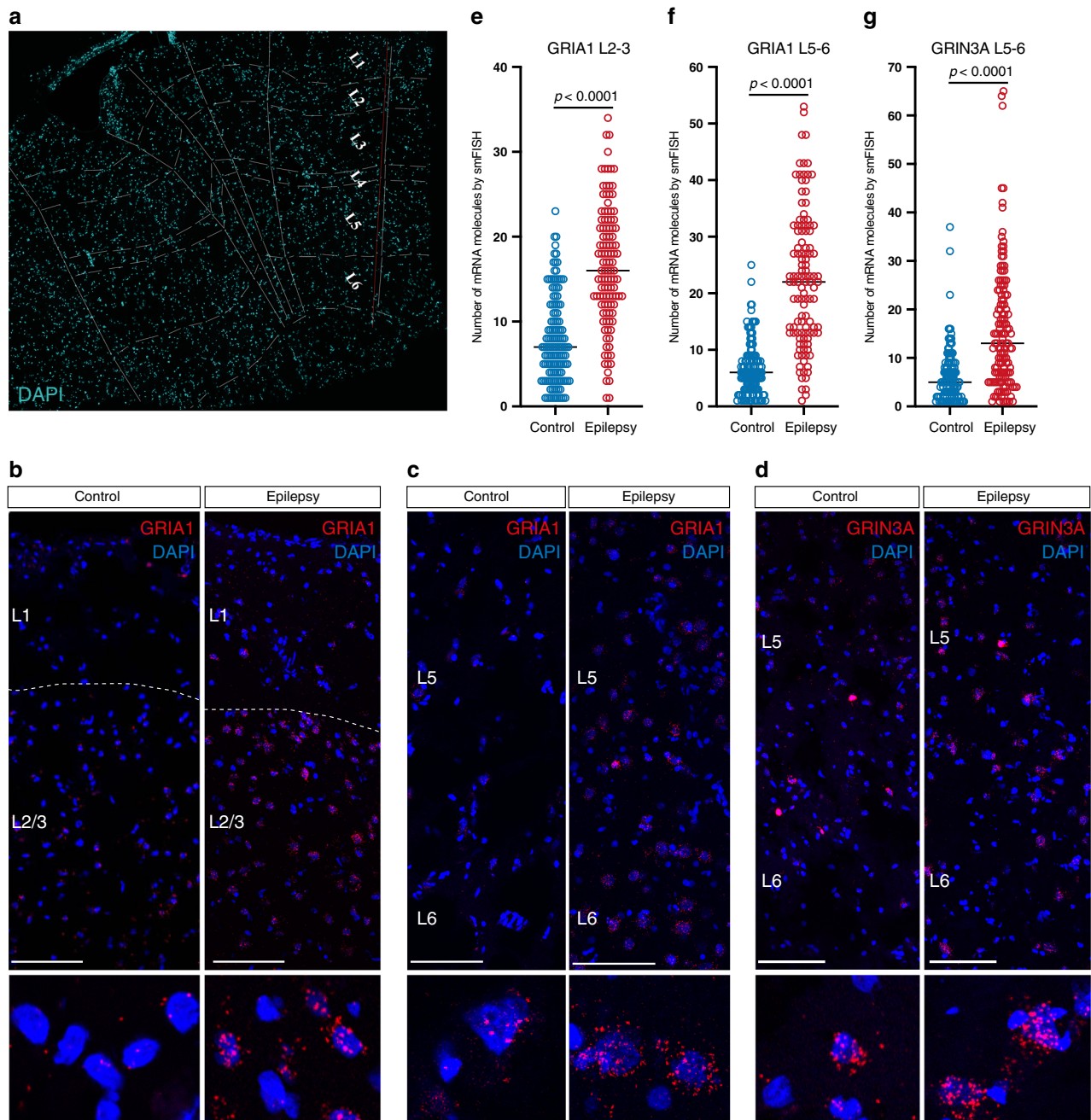

**Fig. 6 Layer-wise dysregulation of gene expression of glutamate receptor subunits in the cortex of epileptic patients. a** Overview of a brain section showing DAPI labeling that was used for identification of the cortical layers. **b–d** Representative images of the temporal cortices of control (nonepileptic) and epileptic brains showing an increased number of mRNA molecules for GRIA1 in L2–3 (**b**), GRIA1 in L5–6 (**c**), and GRIN3A in L5–6 (**d**). **e–g** Quantifications showing the increase of mRNA molecules in the epileptic temporal cortex for GRIA1 in L2–3 (**e**: one-tailed Mann–Whitney test, 136 cells in three sections from three control brains and 113 cells in three sections from three epileptic brains), GRIA1 in L5–6 (**f**: one-tailed Mann–Whitney test, 145 cells in three sections from three control brains, and 117 cells in three sections from three epileptic brains) and GRIN3A in L5–6 (**g**: one-tailed Mann–Whitney test, 130 cells in two sections from two control brains and 143 cells in three sections from three epileptic brains).

Together, these gene co-expression analyses, in light of the other cell-level results, capture core transcriptional networks active particularly in L5–6_Fezf2_Tle4_Abo and L2–3_Cux2 subtypes of principal neurons. Gene modules associated with epilepsy status point toward transcriptional changes related to synapses and ion channels, including AMPA receptor auxiliary subunits, glutamate receptor subunits, and voltage-gated sodium channels, likely to be implicated in aberrant hyperexcitation in epileptic circuits.

**Neuronal circuits that contribute to epilepsy.** To predict neuronal subtypes that have the highest potential to contribute to epilepsy, we integrated data obtained in previous analyses and scored all neuronal subtypes in epileptic temporal cortex based on the scale of total transcriptomic changes and various epilepsy-related changes studied above (see "Methods"). Based on all the data above, we can predict that L5–6_Fezf2_Tle4_Abo, L3_Cux2_Prss12, L2_Cux2_-Lamp5, and L2–3_Cux2_Frem3 subtypes, as well as Vip_Cbln1, Pvalb_Sulf1, Sst_Tac1, and Id2_Lamp5 subtypes are most affected

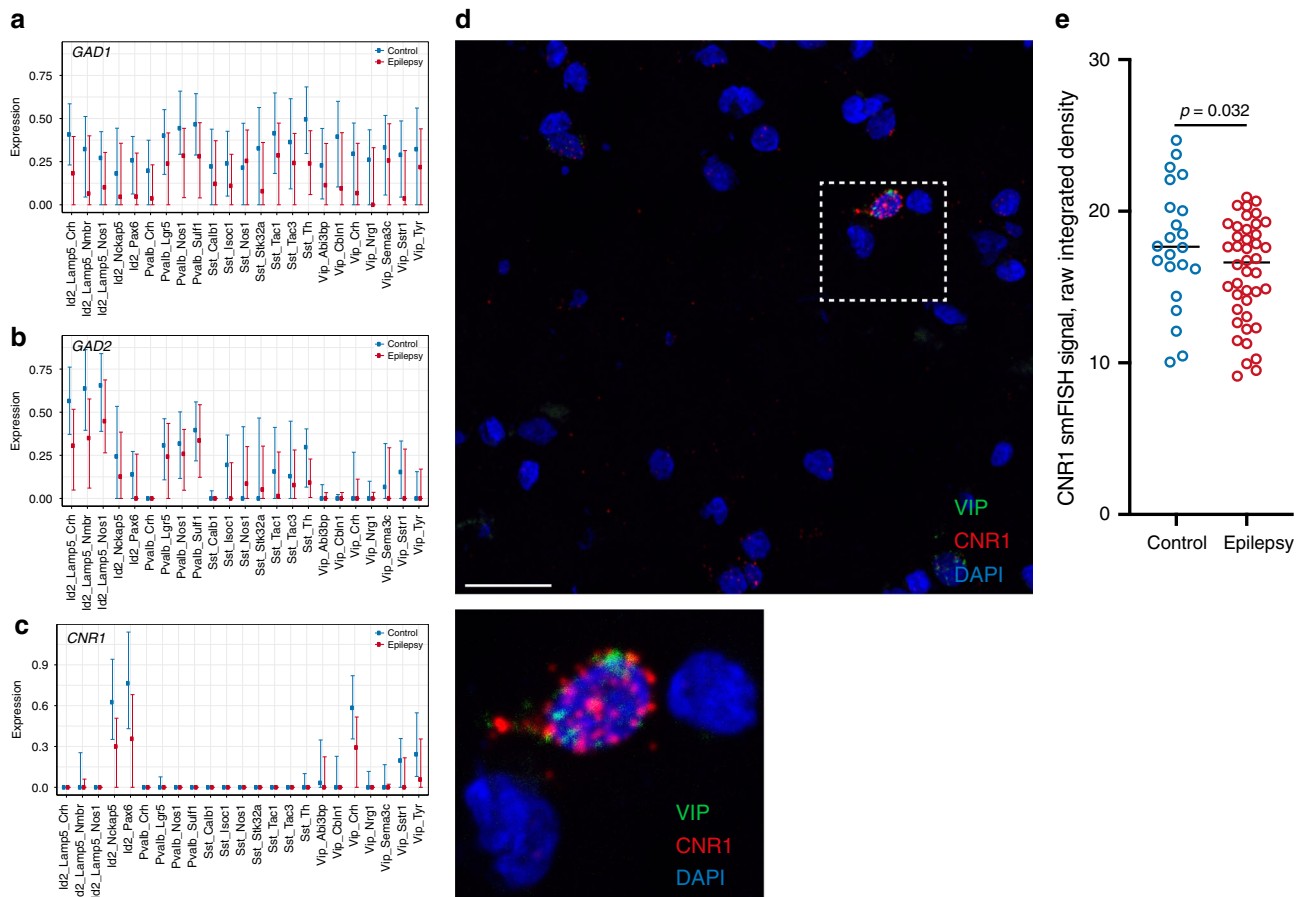

**Fig. 7 Dysregulation of genes underlying hypoinhibition in the cortex of epileptic patients. a, b** Visualization of *GAD1* and *GAD2* expression to assess hypoinhibition of GABAergic interneurons revealed a general decrease in expression of these genes involved in GABA synthesis. Total-count normalized expression level is shown on the *y* axis. Points represent median expression values, averaged over all samples for each subtype. Whiskers show 25 and 75% levels of expression, also averaged over all samples. **c** Decreased expression of cannabinoid receptor 1 (*CNR1*) in Vip and Id2 non-Lamp5 subtypes. **d** Representation image showing smFISH analysis of CNR1 transcript abundance in VIP-positive GABAergic interneurons in a nonepileptic cortex. **e** Quantitative analysis of CNR1 transcript abundance by smFISH demonstrating a decrease in *CNR1* expression in VIP-positive GABAergic interneurons in epileptic cortex (one-tailed unpaired *t* test, 22 cells in three sections from three control brains and 43 cells in five sections from five epileptic brains).

by epilepsy subtypes of principal neurons and interneurons, respectively (Fig. 8a and Source Data Table 20). Thus, it might be that epilepsy-related circuits mainly involve L2–3_Cux2 and L5–6_Fezf2 principal neurons. For interneurons, we observe a rather variable effect of epilepsy on specific subtypes within cardinal classes, indicating selective vulnerability and differential contribution to epilepsy of interneuron subtypes. Interestingly, when we related most affected subtypes identified by integrative analysis with transcriptomic shifts that affect groups of subtypes (Fig. 3d), we noted that most affected subtypes of principal neurons and GABAergic interneurons clustered together according to similarity of the enriched GO terms (Fig. 8b). Thus, L2–3_Cux2 and L5–6_Fezf2 subtypes co-clustered with Sst_Tac1 and Vip_Cbln1 subtypes, whereas L3_Cux2_Prss12 co-clustered with Pvalb_Sulf1, which might underlie local neuronal networks with most strongly affected in the epilepsy transcriptome.

## Discussion

Epilepsy is a complex neurological disorder drastically affecting neuronal circuits by overexcitation and seizure activity. However, there is a lack of knowledge of how individual subtypes of neurons are affected by epilepsy and how each subtype can contribute to epileptogenesis. Such data are necessary in particular for the human tissue, in order to understand disease etiology and discover new targets for diagnostics and treatment. Using a

single-cell transcriptomics approach in the human brain, we identified large-scale changes in the transcriptome of the epileptic cortex that were distributed across multiple neuronal subtypes. Furthermore, we identified those subtypes of principal and GABAergic interneurons as the most likely candidates for contributing to seizure triggering and propagation.

Single-cell transcriptomics is a fast-developing technology to study at high resolution how a pathological condition can affect cellular composition and gene expression in a tissue. So far, gene-expression changes in epileptic brains have been studied by bulk transcriptomics in resected pieces of brain tissue, containing all types of neurons and glia as well as nonneural cells[13–15,17,18,39]. Bulk transcriptomics is prone to high sample-to-sample variability due to the averaging of gene expression across all cell types, which is likely the reason that epilepsy bulk transcriptomics studies identified a rather small set of genes that changed expression in epileptic brains. Our single-cell transcriptomics analysis of the epileptic temporal cortex greatly increases the resolution of gene expression changes in the epileptic cortex. Thus, we discovered that epilepsy is characterized by the dysregulation of thousands of genes with up-/downregulation in specific neuronal subtypes (in total, by summing up dysregulated genes across all subtypes, ~6,900 and 13,700 DE genes for GABAergic and principal neurons, respectively, Source Data Table 6). Furthermore, the majority of dysregulated genes

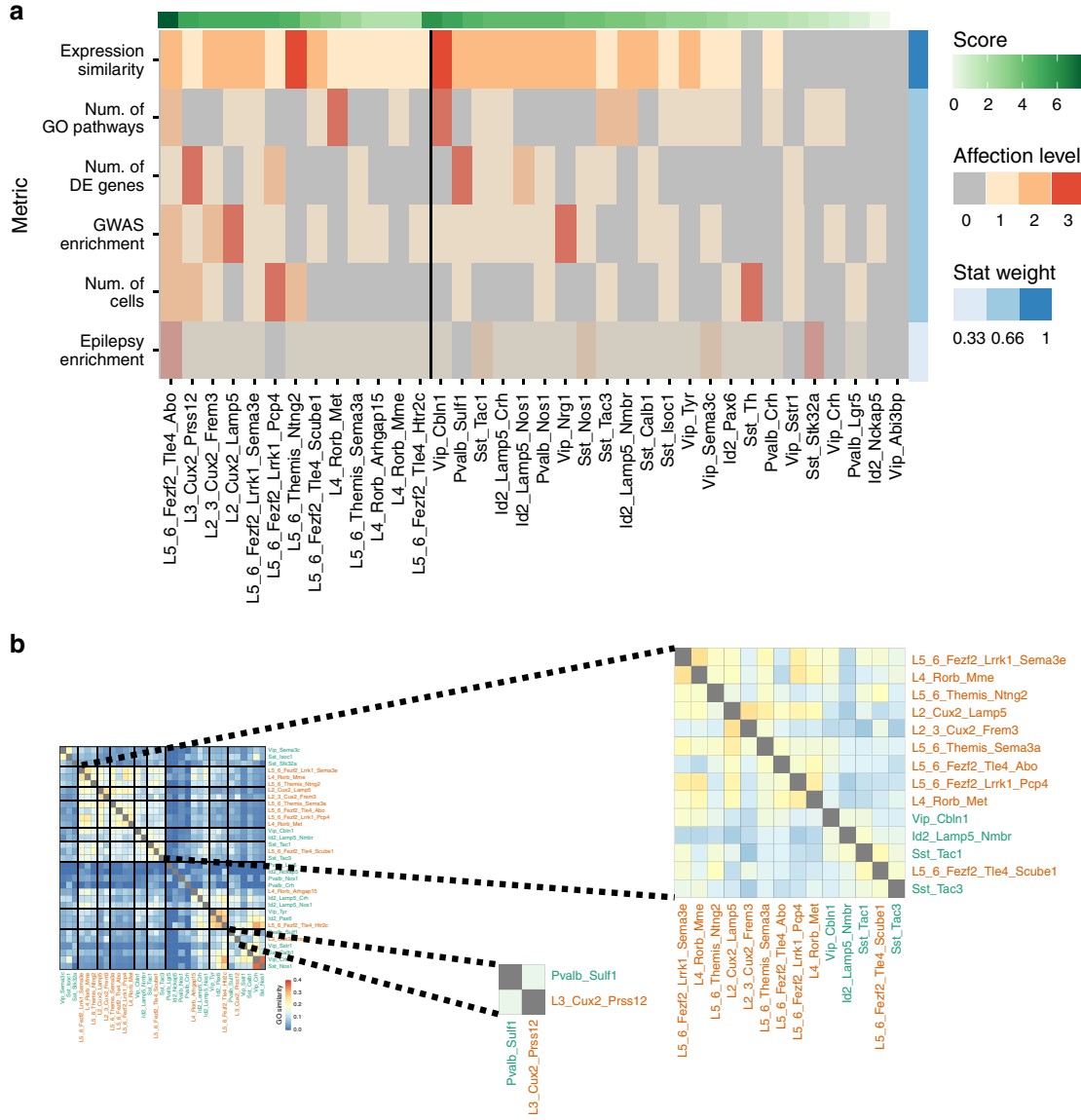

**Fig. 8 Neuronal subtypes most affected by epilepsy based on integrative analysis. a** Six metrics—expression similarity, cell-type composition, number of changed GO terms, enrichment in GWAS genes, enrichment in epilepsy genes, and number of DE genes—are aggregated into a single score. The six metrics are shown on the y axis, ordered by the metric weight, where the weight is represented by the blue vertical colorbar on the right and by the color transparency of the rows. Cell types grouped into interneurons/principal neurons are shown on the x axis, ordered by the total score (green horizontal colorbar on the top). The colors on the heatmap represent the strength of the effect: 0—not affected, 1—affected, 2—highly affected, 3—most affected. For a rational of how we assigned the level of effect and weight for each metric, see "Methods". **b** Clustering of cell subtypes based on enrichment of certain GO terms (similar to Fig. 3d). Enlarged inlet shows co-clustering of cell subtypes with the most affected transcriptomes (based on the metric in Fig. 8a).

identified in our study were not previously reported for epilepsy, and thus are novel epilepsy-associated genes. These genes should be integrated with data from genetic and functional studies to update our knowledge regarding the mechanisms of epileptogenesis[40]. Importantly, based on the number and function of the dysregulated genes, we reveal a large variability in transcriptomic changes in epilepsy between neuronal subtypes, which is likely associated with differences in vulnerability and contribution to seizure activity. In summary, single-cell transcriptomics analysis unraveled a high complexity of gene expression changes in epilepsy, which was not possible to see in bulk transcriptomics studies.

One of the major findings in our study is the discovery that the effects of epilepsy on various neuronal subtypes differ in severity. Previously, involvement of neurons in epilepsy has been mainly studied layerwise for principal neurons or for cardinal types of GABAergic interneurons, e.g., PV, SST, or VIP[5,41,42]. Here, we report that within a single layer of principal neurons and a cardinal type of GABAergic interneurons, there are major differences between epilepsy effect on individual subtypes (Fig. 8a). The effect is most variable in GABAergic interneurons, likely due to the high heterogeneity of this neuronal class. Thus, of four and seven subtypes for Pvalb, Sst, and Vip families of interneurons, respectively, Pvalb_Sulf1, Sst_Tac1, and Vip_Cbln1 exhibit the largest transcriptomic effect in epilepsy. It shows that the disease phenotype should be investigated beyond the classes/cardinal types, thus unraveling how individual neuronal subtypes are affected.

Several principal-neuron subtypes in layers L5–6 had a dramatic enrichment in epilepsy-related transcriptomic changes, and, in particular, dysregulation in gene expression for multiple glutamate receptor- and action potential-associated proteins.

Thus, for the AMPA receptor complex, there was an increase in expression of GRIA1 (GLUR1) and decrease in GRIA2 (GLUR2), as well as upregulation of genes coding for several AMPA auxiliary proteins including CKAMP44 (SHISA9)[34,35]. Such changes in expression of GLUR1 and CKAMP44 likely increase neuronal excitability, due to the increase in the number of GLUR1-containing AMPA receptors and increase in current amplitudes for AMPA receptors[35]. Besides, downregulation of GLUR2 would increase $Ca^{2+}$ permeability that should in turn contribute to alterations in intracellular signaling via $Ca^{2+}$ (e.g., plasticity) and may increase the risk of seizure-induced cell death (excitotoxicity)[43]. Changes in expression of NMDA, kainate, and metabotropic glutamate receptors in different subtypes of L5–L6 neurons of the epileptic cortex are also likely to contribute to overexcitability. For instance, postsynaptic GLUN3A (GRIN3A) substantially alters the function of NMDARs, and is believed to be protective by downregulating synapse number[44]. In addition, an increase in GLUN3A expression was shown to enhance tonic activity of pre-NMDA receptors and evoked neurotransmitter release in the mouse cortex[45].

Sst_Tac1/3 interneurons were among the subtypes most affected by epilepsy across GABAergic interneurons based on the metric that assessed changes in gene expression and cell-type composition. Furthermore, pathway analysis showed that Sst_Tac1/3 has the highest number of GO terms containing epilepsy-related DE genes, which span from neurotransmission to developmental processes. Although there is little human data about the involvement of cortical Sst interneurons in epileptogenesis, in mouse models of epilepsy, Sst interneurons have been shown to contribute to seizure generation and propagation[5,46]. By relating our annotation to previously reported layer-wise location in ref. [7], both Sst_Tac1 and Sst_Tac3 were assigned to L4–6 (Supplementary Fig. 6d), which might indicate high underlying vulnerability of deep-layer circuits to epilepsy.

Another GABAergic interneuron subtype, Pvalb_Sulf1, also exhibited a substantial effect on the transcriptome and the effect was larger compared to other Pvalb subtypes. Pvalb_Sulf1 neurons in the epileptic cortex exhibited increased expression of the AMPA receptor subunit GLUR1 as well as the AMPA auxiliary subunit TARP-γ3. Furthermore, expression of GAD1, the gene coding for the GABA-synthesizing enzyme, was dramatically reduced, indicating decreased availability of GABA for sufficient inhibition of principal neurons. Large remodeling of the neurotransmission-related transcriptome in Pvalb_Sulf1 interneurons could affect their cellular properties, in particular high-frequency firing[47]. Impaired high-frequency firing of parvalbumin interneurons leads to disturbed synchronization of principal-neuron activity and disruption of brain oscillations as previously reported for epilepsy[48].

The results of our study point to a module-like dysregulation of the transcriptomes of human epileptic cortices, where modules represent neuronal assemblies with distinct transcriptomic changes. First, there were clear layer-specific changes in gene expression and signaling pathways. Second, neuronal subtypes clustered together according to changes in gene expression (both for individual genes and for GO terms), which suggests coordinated shifts in the transcriptome across several principal neuron and GABAergic interneuron subtypes. Last, neuronal subtypes with transcriptomes most changed in epilepsy (based on a multimodal metric in Fig. 8a) also clustered together according to the transcriptional similarity of their DE genes (Fig. 8b, overview for the whole clustering in Fig. 3d). Such a module-like organization of epilepsy-affected neuronal assemblies has not been reported before, again emphasizing high power of single-cell transcriptomics to identify disease-related changes.

Interestingly, one of the most dysregulated gene families in our epilepsy samples are AMPA auxiliary subunits[49]. CKAMP44 (SHISA9), TARP-γ3 (CACNG3), TARP-γ4 (CANCG4), cornichon 3 (CNIH3), and GSG1L (GSG1L) were upregulated in one or several principal-neuron subtypes with a characteristic layer-wise upregulation pattern (Fig. 5a). Thus, while CKAMP44 exhibited a dramatic increase in expression across all cortical layers, cornichon 3 and GSG1L exhibited a rather selective upregulation in specific subtypes of principal neurons. GABAergic interneurons also showed upregulation in the expression of AMPA auxiliary subunits. For instance, the expression of CKAMP52 and CKAMP44 was increased in Id2_Lamp5 and Sst_Nos1 subtypes, respectively, whereas TARP-γ2 or 3 were increased in Pvalb_-Sulf1, Sst_Tac3, Id2_Nckap5, and few others. Based on its individual properties, each AMPA auxiliary subunit exhibits specific modulation of AMPA receptor function[33–35], which may have an effect on neuronal properties and function. For instance, synaptic strength may be increased and short-term plasticity affected with an increased number of CKAMP44-bound AMPA receptors[34,35]. Importantly, based on previous findings on the function of CKAMP44[50], a possible consequence may be a more pronounced short-term depression, which could be a homeostatic mechanism to protect neurons in hyperactive network states. In addition, GSG1L decreases conductance of AMPARs[51]. Thus, the upregulation of GSG1L may counteract hyperactivity by decreasing the strength of excitatory synapses. Finally, another major gene set dysregulated in epilepsy was related to action potential. Most of the dysregulated genes are essential for controlling the kinetics of action potential, and thus a change in their expression should modify the properties of the affected neurons.

By categorizing transcriptomic differences between neuronal subtypes in nonepileptic and epileptic cortices, we demonstrated that one of the largest affected gene categories are those associated with neurotransmission. This result is corroborated by multiple studies that showed changes in neurotransmission in epilepsy both at pre- and postsynaptic sites (see refs. [52–54]). Another major category of DE genes in epilepsy is associated with changes in neuronal morphology that could be related to remodeling and connectivity caused by epileptogenesis[55,56]. Interestingly, a number of genes associated with brain development were dysregulated in neuronal subtypes in the epileptic cortex (Supplementary Figs. 12–15). Since all the patients we studied have TLE of neurodevelopmental origin, some of the dysregulated developmental genes discovered in our study could underlie early pathogenesis of epilepsy and trigger formation of epilepsy-related neuronal networks.

In conclusion, we have identified large-scale and complex changes in the neuronal transcriptomes of epileptic patients, where some subtypes showed a dramatic epilepsy-driven dysregulation of gene expression, whereas other subtypes were largely spared. We show that epilepsy-related transcriptomic alterations could be clustered into modules containing several neuronal subtypes that might underlie distinct neuronal assemblies affected by epilepsy. Future translational studies in mouse models and human tissue are necessary to resolve which of the identified pathways leads to seizure generation and propagation, and which rather represent the homeostatic plasticity of neuronal networks. Nevertheless, based on our results, it is likely that antiepileptic therapies should take into consideration the interplay between subtypes and their relationship in circuits for effective treatment and seizure relief.

## Methods

**Sample information**. The description of human brain samples is provided in Source Data Table 1. Epilepsy cortex samples were obtained from temporal lobectomies of patients undergoing surgery for TLE in the Departments of

Neurology and Neurosurgery at Rigshospitalet, Copenhagen. During epilepsy surgery, a part of the temporal cortex is removed to enable access to the underlying epileptogenic hippocampus, which is subsequently removed. The collection of human brain samples has been approved by the Ethical Committee in the Capital Region of Denmark (H-2-2011-104), and written informed consent was obtained from all patients before surgery.

Postmortem brain material was collected by the Human Tissue Brain Bank—Semmelweis University (HBTB). Brain autopsy and use of material and clinical information for research purposes was authorized by the Committee of Science and Research Ethics of the Ministry of Health, Hungary, and the Regional Committee of Science and Research Ethics of Semmelweis University. Permission numbers are 6008/8/2002 and 32/1992/TUKEB.

The parts of the temporal cortex that were dissected during biopsies or autopsies include Brodmann areas 20/21/22/38.

**Nuclei extraction and isolation by FANS.** Human brain tissue from the temporal cortex was collected at the Departments of Neurology and Neurosurgery at Rigshospitalet (Copenhagen) from patients undergoing surgery for drug-resistant TLE. The tissue was either frozen directly on dry ice or collected in cold Hibernate-A medium (Gibco, A1247501) and subsequently frozen on dry ice after collection (time from resection to freezing <1 h for all TLE samples). The collection of human brain samples has been approved by the Ethical Committee in the Capital Region of Denmark (H-2-2011-104), and written informed consent was obtained from all patients before surgery. Nuclei extraction was performed as described before[57] with some modifications. Prior to nuclei extraction, nuclei isolation medium 1 (NIM1) (250 mM sucrose, 25 mM KCl, 5 mM MgCl$_2$, and 10 mM tris buffer, pH 8), NIM2 (NIM1 buffer supplemented with 1 μM DTT (Thermo Fisher Scientific) and 1× EDTA-free protease inhibitor (Roche)), and homogenization (NIM2 buffer supplemented with RNaseIn (0.2 U/μL, Clontech), Suprasin (0.2 U/μL, Invitrogen), and Triton (0.1% v/v)) buffers were prepared. Briefly, the sectioned frozen brain tissue was placed into a precooled Dounce homogenizer with ice-cooled homogenization buffer. The tissue was dissociated on ice using 5–6 strokes with a loose pestle and 15–17 strokes with a tight pestle. The homogenate was first filtered through a 70-μm filter. Nuclei were collected (900 g, 10 min) and blocked on ice for 15 min in blocking solution (RNase-free H$_2$O, 0.5% BSA (w/v)), and RNase inhibitor (0.2 U/μL, Clontech) to minimize unspecific antibody binding. Subsequently, nuclei were stained for mNeuN-488 (1:2000, Millipore MAB377X) for 45 min rotating at 4 °C. To enrich for interneurons, some of the nucleus preparations were co-stained with rbSox6 (1:2000, Abcam, AB30455), mainly in Pvalb and Sst interneurons[58]. Alternatively, a guinea pig Sox6 antibody was used (1:2000, gift from Prof. Dr. Michael Wegner, Institut für Biochemie, FAU Erlangen-Nürnberg). As gating control, mouse IgG1 κ Isotype (1:1000, BD Pharmingen 554121) was used for NeuN sorting and normal rabbit IgG Isotype (1:2000, R&D Systems AB-105-C) for rbSox6 sorting.

Nuclei were washed (500 g, 5 min), resuspended in blocking buffer containing 7-aminoactinomycin (7-AAD) (1 mg/mL, Sigma), and incubated on ice until sorting. Neuronal nuclei were isolated using a FACSAria I (70-μm nozzle) in single-cell mode in 96-well plates containing 2 μL of lysis buffer (1.9 μL of Triton X-100 (0.2%) + 0.1 μL of RNase Inhibitor (40 U/μL)) per well. The multiwell targeting efficiency was assessed by imaging single fluorescent beads (Alignflow™ Flow Cytometry Alignment Beads for Blue Lasers, 2.5 μm), which were sorted using the identical sort mode as used for the human neuronal nuclei. Imaging was performed using ScanR, which is part of the Olympus (2.6.1 version) fluorescent microscope (IX83 Microscope). After collection of single nuclei, the plates were gently vortexed for 15 s and spun at top speed for 60 s (VWR Plate PCR plate spinner). The plates were placed on dry ice immediately and stored at −80 °C until use.

**Smart-seq2 on human neuronal nuclei.** After thawing the plate, 1 μL of dNTP mix (10 mM) and 1 μL of oligo-dT (10 μM, /5Biosg/AAGCAGTGGTATCAACG-CAGAGTAC(T)30VN-3') was added to each well and the lysis protocol was carried out according to ref. [24]. Next, 5.31 μL of reverse-transcription mix containing 0.25 μL of RNase I, 2 μL of 5× first-strand buffer, 2 μL of betaine (5 M), 0.06 μL of MgCl$_2$ (1 M), 0.5 μL of DTT (100 mM), and 0.5 μL of SuperScript III (200 U/μL) was added to each well and reverse transcription was carried out as follows: 42 °C—90:00, 42 °C—hold, 42 °C—12:20, 10 cycles (50 °C—2:00, 42 °C—2:00), 39 °C—12:00, 70 °C—15:00, 4 °C—hold. After the first 90 min of RT, 0.4 μL of TSO (0.5 μM final concentration) was added to each reaction at room temperature. The plate was resealed, centrifuged for 1 min at 750 g, and placed back on the thermocycler to resume cycling from 42 °C—12:20. After RT, 15 μL of cDNA enrichment PCR reaction mix containing 12.5 μL of 2× KAPA HiFi HotStart ReadyMix, 0.25 μL of ISPCR primer (10 μM, /5Biosg/AAGCAGTGGTATCAACGCAGAGT-3'), and 2.25 μL of nuclease-free water was added to each well and cDNA enrichment PCR was performed as follows for 18 cycles (modified after[59]): 98 °C—3:00, six cycles (98 °C—0:20, 60 °C—4:00, and 72 °C—6:00), six cycles (98 °C—0:20, 64 °C—0:30, and 72 °C—6:00), and six cycles (98 °C—0:20, 67 °C—0:30, and 72 °C—7:00), 72 °C—10:00, 4 °C—hold. The cDNA purification was performed as described previously[24] using either AMPure XP or SPRI beads at a ratio of 1:1 (both Beckmann). For elution, Illumina Nextera XT resuspension buffer or Qiagen elution buffer (EB) was used. cDNA concentrations were measured using the Qubit 3.0 fluorometer

according to the manufacturer's protocol, and selected cDNA samples analyzed on Agilent 2100 Bioanalyzer using the HS DNA assay.

**RNA-sequencing library preparation for Smart-seq2 using Nextera XT.** Dual-indexed Illumina Nextera XT sequencing libraries were prepared using 25–50% of the recommended reaction volumes of Nextera XT components. All Nextera XT libraries were prepared using 350 pg of input cDNA for tagmentation, and were subsequently enriched for 12 PCR cycles, and purified using either AMPure XP or SPRI beads (both Beckmann) at a sample:bead ratio of 0.6:1. Library concentrations were measured using Qubit 3.0 fluorometer, and the fragments were analyzed on the Agilent 2100 Bioanalyzer using the HS DNA assay. All libraries were individually normalized and diluted to a final concentration of 2 nM. The equimolar pooled (2 nM) single-cell libraries were denatured and diluted to a concentration of 1.6–1.7 pM and sequenced on Illumina NextSeq500 in a single-end 75-bp format (FC-404-2005, Illumina). Single-nuclei libraries were sequenced at an average depth of 3mio reads.

**10× Chromium on human neuronal nuclei.** Human neuronal nuclei were isolated in bulk (FACS isolation buffer: PBS (1×) (Gibco/Ambion) + 0.04% BSA (Ambion) + 0.2 U/uL RNAse Inhibitor (Clontech/Takara)) by flow cytometry as described for Smart-seq2 using either FACSAria I or III (70-μm nozzle), based on NeuN expression. The entire isolated NeuN+ nuclei fraction per brain sample was loaded on the 10× Chromium chip to perform the Single Cell 3′ v2 and v3 chemistry according to the manufacturer's instructions. Samples with less NeuN+ nuclei were diluted using the FACS isolation buffer instead of RNase-free H$_2$O. The NeuN-nuclei fraction from EP4 and CNT5, 6, and 8 brains was used for sequencing of nonneuronal nuclei. For enrichment PCR, 12 PCR cycles were applied (except for E2, where we applied 16 cycles due to low input). For the Illumina library preparation, 10–14 cycles of enrichment PCR were performed. Illumina libraries were diluted to a concentration of 2 nM as described for Smart-seq2 libraries, and the libraries were pooled for sequencing. Following denaturation, the library pool was diluted to a final loading concentration of 1.6–1.7 pM and sequenced on a NextSeq500 or diluted to 400 pM loading concentration and sequenced on the Novaseq6000 in paired-end mode (for NextSeq500 10X v2: read 1—26 cycles, read 2–98 cycles, index 1–8 cycles; for NextSeq500 10X v3: read 1–28 cycles, read 2–91 cycles, index 1–8 cycles; for Novaseq600 10X v3: read 1–28 cycles, read 2–94 cycles, index 1–8 cycles) using single-index read at an average depth of 16,000–50,000 reads per nucleus.

**In situ hybridization by RNAscope.** In situ hybridization (ISH) was performed according to the manufacturer's instructions (ACD—RNAscope Multiplex Fluorescent Reagent Kit v2 (Cat No. 323100)) with minor modifications. Thus, the protease treatment was increased to 45 min by adding fresh protease for 15 min after the standard incubation of 30 min. In addition, the slices were incubated for 30 s at room temperature with TrueBlack Lipofuscin Autofluorescence Quencher (Biotum, Cat No. 23007). This step was done before the DAPI staining. The probes were purchased from ACD:

Cnr1-C1 (Cat No. 591521), Shisa9-C1 (custom-made), Gria1-C1 (Cat No. 472441), Grin3a-C3 (Cat No. 534841-C3), Vip-C2 (Cat No. 452751-C2), and Rorb-C2 (Cat No. 446061-C2). The probes were visualized with different fluorophores, namely Opal-570, Opal-520, or OpaI-690 (Akoya, FP1488001KT & FP1487001KT & FP1497001KT).

All images were acquired using a SP5 Confocal microscope at the IMB Microscopy Core Facility, Mainz, Germany (20× objective, 0.7 NA). Image sizes are either 4096×4096 pixels or 2048 × 2048 pixels. For image preprocessing, we used ImageJ[60].

The analysis for *GRIA1*, *GRIN3A*, and *CKAMP44* (*SHISA9*) is based on dots counted per cell. We analyzed the expression of *GRIA1* and *GRIN3A* in cortical layers 2–3 and 5–6. Double-FISH experiments were performed to analyze *CKAMP44* expression specifically in *RORB*-positive cells.

Due to a high number of dots labeling for CNR1 mRNA, the dots were overlapping in many Vip-positive cells, and an unequivocal quantification of individual dots were therefore not possible. Instead, we quantified the integrated intensity of the CNR1 signal per Vip-positive cells normalized to the Vip signal intensity (*VIP* expression was similar for control and TLE samples based on snRNA-seq data). Images were acquired with the same laser setting. Further analysis was performed in R. Briefly, the background-subtracted integrated intensity was log2 transformed, and a quantile normalization was performed on the VIP signal using the R package 'preprocessCore'[61]. The CNR1 signal was multiplied with a correction factor based on the VIP signal (see Eq. (1)). This normalization was necessary due to variability in signal intensity of FISH that could arise due to several factors, including tissue integrity for FISH, tissue processing etc. To avoid Infinite values, we used Raw Integrated Density instead of Integrated Density. A Welch Two Sample one-sided test was used based on unequal variances in the distributions.

$$\text{cnr1}_{\text{normalized}} = \text{cnr1}*(\text{vip}_{\text{normalized}}/\text{vip}). \tag{1}$$

In total, the following number of samples (*N*) and cells (*n*) was used: CNR1/VIP ($N_{\text{control}} = 3$; $n_{\text{control}} = 22$; $N_{\text{TLE}} = 5$; $n_{\text{TLE}} = 43$); CKAMP44/RORB ($N_{\text{control}} = 2$; $n_{\text{control}} = 98$; $N_{\text{TLE}} = 3$; $n_{\text{TLE}} = 215$); GRIA1 in layer 2–3 ($N_{\text{control}}=3$; $n_{\text{control}} = 136$;

$N_{TLE} = 3$; $n_{TLE} = 113$); GRIA1 in layer 5–6 ($N_{control} = 3$; $n_{control} = 145$; $N_{TLE} = 3$; $n_{TLE} = 117$); GRIN3A ($N_{control} = 2$; $n_{control} = 130$; $N_{TLE} = 3$; $n_{TLE} = 143$).

**Processing of Smart-seq2 single-nucleus transcriptomes.** Fastq files were trimmed with Trimmomatic 0.36[62] with parameters HEADCROP:12, LEADING:3, TRAILING:3, SLIDINGWINDOW:4:15, and MINLEN:25. Trimmed reads were aligned to the GRCh38.p12_genomic.fna genome assembly (accessible from https://www.ncbi.nlm.nih.gov/assembly/GCF_000001405.38) with STAR 2.5.3a[63]. On average, approximately 90% of reads were uniquely mapped to the reference genome. Genes were counted with featureCounts 1.5.1[64] using GRCh38.p2 updated by removing duplicate Entrez gene entries from the gtf reference. Cells with less than 2000 genes were excluded. Genes present in less than five cells were excluded. Cells with more than one mio reads or more than 5% mitochondrial genes were removed. In addition, based on overlapping expression of cardinal markers for neuronal cells, astrocytes, and oligodendrocytes, 150 nuclei (constituting 15% of the total dataset) were removed from further analysis as they could not be assigned to one of the main neural cell types and thus likely represent technical artifacts such as doublets.

Nuclei that remained after filtering were normalized, scaled, and embedded in two-dimensional space with Pagoda 2 package (https://github.com/hms-dbmi/pagoda2 and[65]) using 1000 overdispersed genes and 100 principal components.

**Genome alignment and quality control of 10× Chromium single-nuclei transcriptomes.** Raw data were demultiplexed, aligned, and quantified using Cell Ranger version 3.1 using a custom version of the GRCh38-1.2.0 human reference genome provided by Cell Ranger where "transcript" rows were changed to "exon" in order to allow for the counting of introns (awk 'BEGIN{FS = "\t"; OFS = "\t"} $3 == "transcript"{ $3 = "exon"; print}' \refdata-cellranger-GRCh38-1.2.0/genes/genes.gtf > GRCh38-1.2.0.premrna.gtf). To filter cells, we used "#UMIs * #Cells" versus "log10(#UMIs)" plots, provided by dropestr package.[66] Afterward, cells for which the total mitochondrial gene expression exceeded 8% were removed. Doublets were filtered using the Scrublet package[67], followed by removal of clusters with double-cell-fate signatures (Source Data Table 2).

**Joint analysis of 10× Chromium-processed single-nuclei datasets.** After the initial filtration, each of the 19 NeuN+ datasets (ten healthy and nine epilepsy) was processed with Pagoda 2 using 1000 overdispersed genes and 100 principal components. Afterward, the Pagoda objects were aligned using Conos with parameters $k = 15$, k.self = 5, and k.self.weight = 0.1. The resulting graph was embedded with the UMAP embedding and clustered using the leiden method with Resolution = 15. Cells from the whole dataset were assigned to one of the major cell types based on the expressed marker genes (see Source Data Table 3). After that, clusters expressing a combination of nonneuronal markers, as well as clusters with mixed identities were removed, and the remaining cells were annotated to the final depth. For better alignment, we reran Conos on the filtered data using balancing across conditions (parameter balancing.factor.per.sample) and setting parameters $k = 50$, k.self = 5, k.self.weight = 0.1, k.same.factor = 5, and same.factor.downweight = 0.25. The final UMAP embedding was generated with parameters n_epochs = 1000, spread = 5, min_dist = 1.0.

**The NeuN dataset was processed in the same way, but without Conos alignment.** To estimate the effect of epilepsy on the cell types and for Gene Ontology analyses, we performed additional filtration of samples with a high fraction of missed cell types. We filtered all samples in which >5 subtypes had <5 nuclei, as without the filtration, these samples biased subsequent analyses for the subtypes with low number of cells. Having low number of cells in some samples can compromise the analysis as a consequence of possible imprecisions in the annotation or higher variance of the centroid estimation for certain subtypes. Thus, we removed samples with low number of nuclei per sample (as in samples C3 and E5) or with low-quality sample preparation (sample C5). After filtering, we ran Conos using the same parameters as above.

**Joint analysis of Smart-seq2 and 10× data.** We used Conos label transfer routines to transfer Smart-Seq2 cell-type annotations to our 10× data. We used the preprocessed Pagoda object containing the Smart-Seq2 data and aligned it with the preprocessed 10× Pagoda objects using the same parameters as above ($k = 15$, k.self = 5, and k.self.weight = 0.1). The UMAP embedding was estimated with parameter spread = 1.5 and min.dist = 1, and propagateLabels function with "max.iters = 50".

To map our annotation to the Allen Brain Institute (ABI) subtypes, we also preprocessed and aligned their Smart-Seq2 dataset. Preprocessing was done using Pagoda 2 in the same manner as for 10×. For the alignment, we used the three largest autopsy control datasets (C6, C7, and C8) and ran them with parameters k = 20, k.self = 5, k.self.weight = 0.1, space = "CCA", same.factor.downweight=0.1, and balancing.factor.per.sample corresponding to the protocol.

To produce alluvial diagrams for matching of cell types, we estimated the number of cells $n_{i,j}$ for each of our cell types $i \in types(ours)$ getting labeled by a specific ABI cell type $j \in types(ABI)$. All occurrences with $n_{i,j} = 1$ were filtered out,

as well as

$$n_{i,j} : \frac{n_{i,j}}{\sum_{k \in types(ABI)} n_{i,k}} < 0.05. \qquad (2)$$

**Estimation of expression-similarity score across conditions.** To estimate how severely each of the subtypes was affected by epilepsy, we developed a metric for expression similarity across conditions. We generated a joint count matrix by row-binding individual total-count normalized count matrices. On the joint matrix, we estimated PCA reduction using 100 principal components. Next, we found the cell-type centroids $\bar{v}_{t,s}$ in this PCA space for each sample $s$ and cell type $t$. For each cell type $t$, we estimated all pairwise Pearson correlations between samples

$$c_{t,s_i,s_j} = cor(\bar{v}_{t,s_i}, \bar{v}_{t,s_j}). \qquad (3)$$

As a similarity score, we used the measure of how far the epilepsy samples were from the control samples, accounting for the cross-sample variation within control samples. To do so, we estimated average pairwise correlations between control samples using the 40% trimmed mean

$$m_t = TM_{0.4}(\{c_{t,s_i,s_j} : s_i, s_j \in control\}), \qquad (4)$$

and the deviation from the mean using median absolute deviation measure

$$\sigma_t = MAD(\{c_{t,s_i,s_j} : s_i, s_j \in control\}). \qquad (5)$$

For each pair of an epilepsy and a control centroid $c_{t,s_i,s_j} : (s_i \in control, s_j \in epilepsy)$, we estimated their difference from the control mean, normalized by deviation within the control datasets using a Z-score-like approach:

$$z_{t,s_i,s_j} = \frac{c_{t,s_i,s_j} - m_t}{\sigma_t}. \qquad (6)$$

The obtained measure is <0 for all transcriptomes that are divergent between conditions, and it is around 0 for the cases where the transcriptional profile is not affected by condition. Consequently, as the epilepsy cell types become more similar to the controls, $z_{t,s_i,s_j}$ grows. As a measure of how much a cell type is affected by epilepsy, we used the distribution of scores $z_{t,s_i,s_j}$ over samples $s_i, s_j : (s_i \in control, s_j \in epilepsy)$ for each of the types $t$.

**Differential expression and Gene Ontology enrichment testing.** To inspect functional differences in the data, we performed Differential Expression (DE) and GO analyses using the following procedures. First, on the Conos object with filtered samples, we found DE genes between conditions for each of the cell types using the Conos wrapper of DESeq2 package[25,68]. Among the found genes, we picked those with absolute Z score >3. Then, for each cell type, we kept only the genes with distinct expression level of raw expression >1 UMI in at least 5% of the cells.

We validated the relevance of the found DE genes according to existing knowledge. For each cell type, we tested the enrichment of its DE genes among (i) genes found in GWAS data, and (ii) published epilepsy-related genes. To calculate the enrichment, we used Fisher exact test, using the union of all expressed genes (see above) across all cell types as background.

Next, we performed GO enrichment analysis with the enrichGO function from the clusterProfiler[69] package using Benjamini–Hochberg false-discovery rate adjustment with $P$-value threshold of 0.05. To avoid autopsy-related pathways, we filtered all terms, for which >20% of the enriched genes belong to the list of autopsy-associated genes published by the Allen Brain Institute[7]. To aggregate the terms that were identified based on the same genes, we performed clustering of the terms by genes, collapsing those with highly similar genes. Thus, we first identified clusters of individual pathways for each subtype using Jaccard distance on the sets of enriched genes (R functions hclust and cutree with parameter $h = 0.66$). Then, for each pair of pathways ($P1$, $P2$), we found all cell types that had both $P1$ and $P2$ enriched and estimated fraction of cell types that assigned $P1$ and $P2$ to different clusters:

$$f_{P_1,P_2} = \frac{\left| cluster_i(P1) \neq cluster_j(P1) \right|_{i,j \in enriched(P1,P2)}}{|enriched(P1, P2)|}. \qquad (7)$$

This fraction was used as a distance metric for hierarchical clustering (R functions hclust with parameter method="average" and cutree with parameter $h = 0.66$). This reduced the number of pathways from 446 to 186, which improved visualization and simplified analysis.

**GO visualization.** We used a heatmap of log $P$ values to visualize the pathway clusters. First, we determined the name of each cluster by picking the name of the pathway with the least mean log $P$ value across cell types across all pathways from this cluster. Then, we built a matrix of minimal log p values for each of the cell types (columns) and each of the pathway clusters (rows). This matrix was clustered by pathway clusters using hierarchical clustering with L1 distance over row-normalized log $P$ values (R functions hclust with parameter method="ward.D" and cutree with parameter $h = 2.5$). According to that, pathway clusters that were

enriched in the similar cell types were grouped together, and groups of pathway clusters with size of at least five were picked for further analysis.

To visualize relationships between pairs "cell-type: pathway cluster" we embedded these pairs in 2D space using UMAP. For each such pair, we picked all genes, enriched in the pathways of the given cluster within the given cell type, and used them to characterize the pair. We estimated the pairwise Jaccard distances on these gene sets and used the resulting distances to pick k=10 nearest neighbors for each pair and pass it to the UMAP embedding (parameter spread=1.5, min_dist=0.2).

To understand which cell types are affected by epilepsy in a similar way, we represented each cell type as a set of pathways, which are enriched in this type. We then estimated weighted Jaccard distances and showed them on a clustered heatmap. The weighting was used to account for those pathways that were detected based on similar sets of genes within one cell type. The weight of the pathway $i$ within the cell type $k$ was estimated as

$$w_{k,i} = \left( \sum_{j \in pathways(k)} JS(genes_{k,i}, genes_{k,j}) \right)^{-1}, \tag{8}$$

where $JS$ is nonweighted Jaccard similarity, "$genes_{k,j}$" is the set of genes for the pathway $k$, enriched within the type $j$, and "$pathways(k)$" are all the pathways, enriched for the type $k$. For the visualization, we clustered these weighted Jaccard distances using hierarchical clustering ($h = 1.2$ for the R cutree function).

**Summary score for the degree at which cell types are affected.** For the results summary, we used six different metrics (expression similarity, cell-type composition, number of changed GO terms, enrichment in GWAS genes, enrichment in epilepsy genes, and number of DE genes, see below), aggregated into a single score. We replaced continuous values with ordinal ones to make the metrics comparable to each other. Thus, for each metric, we classified each cell type into one of the following categories: "not affected", "affected", "highly affected", and "top-1 affected cell type". The last, which by definition includes only one cell type, is assigned separately to Excitatory and Inhibitory neurons. The final score is assigned based on a weighted sum of ranks of the categories, where ranks are integers from 0 to 3. Weights were determined a priori according to our trust in specific metrics (see explanations below).

**Expression-similarity score.** The expression-similarity score is a direct measure of transcriptional change between conditions (Fig. 2e), where a value of zero means that transcriptional profiles are high in similarity across conditions, and a more negative value indicates a lower similarity. For each subtype, we calculated a distribution of scores across all pairs of datasets. The cell types with a score where the upper quartile <0 are labeled as "affected". The "affected" types with median score below median of all medians across the cell types (separately for Excitatory and Inhibitory) are labeled as "highly affected". The type with the lowest median value is the "top-1 affected cell type". As the direct measure, expression-similarity score has weight 1.0.

**Changes in cell-type composition.** The proportion of cells of a specific cell type (Fig. 2d) varies between datasets, as cells are subsampled from the total pool. In addition, we expect that abundance of some cell types is affected by epilepsy. To measure differences in cell-type proportions between conditions accounting for variance within cell type, we used a permutation-test $P$ value. Here we do not perform binary hypothesis testing, but use p values as a continuous measure instead. In particular, as we have relatively few samples with large variance, the power of the permutation test is low and the $P$ values are relatively large. Thus, we labeled a cell type as "affected" if it had a $P$ value <0.2, "highly affected" if $P$ value <0.05, and the "top-1 affected cell type" is the one with the smallest $P$ value. As a change in the abundance of cell types should have a large effect on the system, this metric has weight 0.66.

**Enrichment of genes identified by GWAS.** To evaluate the enrichment of genes identified by the largest GWAS study of epilepsy patients[28] (Fig. 2g), we used Fisher test statistics $h$ as the metric. We cross-compared the DE gene lists for each pairwise comparison (i.e., subtype $x_{epilepsy}$ vs. subtype $x_{healthy}$) with the GWAS list. Cell types with $h > 1$ were labeled as "affected", and those with lower confidence interval for $h > 1$ were labeled as "highly affected". The subtype with the largest $h$ value is designed as "top-1 affected cell type". Gene mutations are likely to cause phenotypical changes, but since the size of the patient cohort in the epilepsy GWAS is not large enough and there is a significant variability in patient diagnosis in the GWAS cohort, it is expected that many epilepsy-related genes are still missed in the GWAS data, and thus we assigned a lower weight of 0.66 to this metric.

**Enrichment of epilepsy-related genes.** For this metric (Fig. 2f), we used the same definitions as in the GWAS enrichment score. Since the number of genes that has been associated with epilepsy is influenced by prior knowledge in the literature, it does not provide us with a reliable measure of the real level at which a cell type is affected; thus, we used a weight of 0.33 for this score.

**Number of highly expressed DE genes, adjusted by the number of cells per cell type.** The number of expressed DE genes linearly depends on the number of cells (Supplementary Fig. 11c). Thus, we performed robust linear regression (R function MASS::rlm) of the number of DE genes $y_{DE}$ by the number of cells

$$x_{cells} : y_{DE} = a * x_{cells} + b. \tag{9}$$

Then we used residuals of the regression ($y_{DE}^{observed} - y_{DE}$) as a measure of how affected a cell type was. Cell types with positive residuals (i.e., above the regression line) are marked as "affected", and those with values >75 percentile among the affected types are marked as "highly affected". The cell type with the largest residual value is the "top-1 affected cell type". The number of DE genes is an important factor, but the linear dependency makes it weakly reliable, as the residuals can be explained by noise. Thus, a weight of 0.66 was assigned to this metric.

**Number of enriched GO pathways, adjusted by the number of highly expressed DE genes.** The number of enriched GO pathways was utilized as another metric of functional changes in cell types, although it linearly depends on the number of DE genes (Supplementary Fig. 11a). We used a similar procedure for this metric calculation as we used for the number of highly expressed DE genes (above): residuals of the robust linear regression of the number of enriched GO pathways by the number of highly expressed DE genes. The weight was similarly set to 0.66.

**Gene filtering for rWGCNA.** The Seurat R package (version 3.1.2)[70] was used for preparing the expression data for co-expression analysis. The cells were split by the second level of annotation (Supplementary Fig. 5a) in order to ensure sufficient cell numbers and expression variation within each subset for the detection of gene co-expression. Genes expressed in fewer than 20 cells in a cluster were removed. Principal component analysis was carried out using the RunPCA function after centering and scaling the data with the ScaleData function, to find 120 principal components (PCs). Genes were then ranked by their highest absolute loading value on any given PC, and the top 5000 genes within each cell cluster were selected for co-expression analysis. Subsequent analyses were performed on the entire dataset, except for CNT9 and CNT10.

**rWGCNA adjacency and topological overlap matrix computation.** Robust Weighted Gene Co-expression Analysis was carried out using the WGCNA R package (version 1[38]). The pickSoftThreshold function was used to identify soft thresholding powers as follows: powers corresponding to the top 95th percentile of network connectivity or above were discarded, and the lowest soft-threshold power between 1 and 30 to achieve a scale-free topology R-squared fit of 0.93 was selected; if none did so, the thresholding power with the highest R squared was used.

In order to identify gene networks robust to outlier cells, the expression data were resampled using a previously published approach[37], drawing two-thirds of the cells at random without replacement 100 times. The consensusTOM command was then run with a consensusQuantile of 0.5, "pearson" correlation coefficient, and "signed hybrid" networkType, to compute a signed consensus Topological Overlap Matrix (TOM). Genes were subsequently filtered with the goodGenesMS output produced by consensusTOM.

**Clustering and intramodular connectivity.** The consensusTOM matrices were converted to distance matrices, and the hclust function was used with the "average" method to cluster genes hierarchically. The cutreeHybrid command was used with a deepSplit of 2 and pamStage set to TRUE to cut the dendrogram into discrete modules, each containing a minimum of 15 genes. We next computed the intra-modular connectivity, or "kIM", of every gene with respect to each module, to serve as a continuous and weighted measure of module membership. Modules whose gene kIM scores exhibited a Pearson correlation of 0.85 or higher were merged. Inspired by[71], kIM scores were then used as distance measures in a subsequent iterative k-means clustering, in which genes were reassigned if their kIM with regard to another module was 1.25 times greater than the kIM to their current module. kIM scores were recomputed for the new modules, and the algorithm was repeated until no further genes were reassigned. Finally, a t test was performed to prune genes whose kIM with respect to their allocated module was not statistically significant (using the Benjamini–Hochberg false-discovery rate adjustment).

**Post-rWGCNA gene module filtering.** Of the original 140 rWGCNA modules detected, those for which 75% or more of the constituent genes were also found with another larger module, with at least a weighted Pearson correlation (using the WGCNA::cor function) between gene kIM scores of 0.75 or higher, were removed. The remaining 129 modules were subsequently filtered down to 117 by removing modules associated with genes differentially expressed in neurosurgery and post-mortem interval conditions[7]. The test was carried out by computing the dot product of module membership scores (kIMs) with –log10-transformed differential expression p values and evaluating significance against a null distribution produced by permuting kIM gene labels (10e3 replicates). A module expression matrix was produced by scaling kIM gene weights to sum to 1 and computing module expression as the weighted sum of module-normalized expression. To address bias

from sample-specific expression profiles, fixed-effect linear models with either sample or epilepsy condition as covariates and module as the outcome were used to remove modules for which the R squared of any sample covariate was higher than that of the epilepsy condition covariate model within the cell cluster from which the module originated, narrowing the field to 38 candidate modules. Confidence intervals and P values were then computed for the epilepsy condition coefficients, resulting in 12 modules with a significant epilepsy status coefficient. Information on all of the original 140 gene modules detected across 8 cell subsets can be found in Source Data Table 14. Regression coefficients, confidence intervals, and P values for simple linear models with epilepsy condition as covariate and module expression as outcome can be found in Source Data Table 15.

**Gene set enrichment testing**. To address potential confounding by common co-expression structures within the gene modules and the curated epilepsy gene set, the correlation of the epilepsy gene expression profile within our single-cell expression data was used to compute a variance inflation factor (VIF)[72]. The VIF was then passed to the rankSumTestWithCorrelation command from the limma R package (version 3.38.3)[73] to carry out a nonparametric Wilcoxon signed-rank testing whether genes from the curated list were ranked near the top of module gene membership scores (kIMs), highlighting seven gene modules after adjusting p values for multiple testing (Bonferroni, 12 tests). Wilcoxon Rank Sum test enrichment results for 12 modules with uncorrected P values (one-sided) can be found in Source Data Table 18.

**Functional module annotation**. To investigate associations of the 7 prioritized gene modules with biological pathways, the gprofiler2 R package (version 0.1.8)[74] was used to query the Gene Ontology Biological Process, Molecular Function, and Cellular Component databases, using Bonferroni correction for multiple testing (Source Data Table 19).

**Module preservation in cell-level-4 subtypes**. Having identified gene modules within subsets of the cells corresponding to the second level of annotation, we used the WGCNA::modulePreservation function to evaluate module preservation within the level-4 subtypes (Supplementary Fig. 5a), using as reference the level-2 cell types, in which the modules were originally detected[75].

**Reporting summary**. Further information on research design is available in the Nature Research Reporting Summary linked to this article.

## Data availability

The RNAseq data are available at EGA, EGAS00001002882. Source data are provided with this paper.

## Code availability

Code to reproduce the analysis is available on git (https://github.com/khodosevichlab/Epilepsy19).

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

## Acknowledgements

We thank BRIC's single-cell core facility, Anna Fossum from BRIC's flow-cytometry core facility, and Yasuko Antoku from BRIC's imaging core facility for technical assistance and advice. We thank Bo Jespersen (Neurosurgery, Rigshospitalet) for surgical resection of epilepsy biopsies. We are grateful to Prof. Dr. Michael Wegner (Institut für Biochemie, FAU Erlangen-Nürnberg) for providing a guinea pig Sox6 antibody. The work was supported by Novo Nordisk Hallas-Møller Investigator grant (NNF16OC0019920), Lundbeck-NIH Brain Initiative grant (2017-2241), DFF-Forskningsprojekt1 (8020-00083B) to K.K., and German Research Foundation (DFG) grant within the Colla-borative Research Center (SFB) 1080 "Molecular and Cellular Mechanisms of Neural Homoeostasis" to J.v.E. U.P. was supported by Lundbeck postdoctoral fellowship (2017-820). V.P. is funded by a cooperative agreement between University of Copenhagen and Harvard Medical School. T.H.P. acknowledges the Novo Nordisk Foundation (NNF18CC0034900) and the Lundbeck Foundation (R190-2014-3904) grants. I.A. acknowledges the Institutional Excellence in Higher Education Grant (FIKP, Semmelweis University). L.H.P. acknowledges Lundbeck Foundation.

## Author contributions

K.K. and U.P. conceived the study. U.P. optimized tissue processing, flow cytometry, and snRNAseq. U.P., M.B., and A.A.M. performed 10× Genomics snRNAseq. U.P. and A.T. performed Smart-seq2 snRNAseq. S.D. and V.P. analyzed snRNA-seq data under supervision of P.K. and K.K. J.M. performed smFISH under supervision of J.v.E. N.V. prepared tissue for smFISH. J.T. performed WGCNA analysis under supervision of T.H. P. L.H.P., J.M., and I.A. obtained brain tissue and patient-related data. L.H.P. coordinated cooperation with Danish Epilepsy Surgery Program. K.K. supervised the study. K.K., U.P., S.D., and V.P. wrote the paper. All authors edited the paper.

## Competing interests

The authors declare no competing interests.
