## [Peer Review File · Nature Communications]

Reviewers' comments:

Reviewer #1 (Remarks to the Author):

The manuscript submitted to Nature Communications, entitled "Identification of epilepsy-associated neuronal subtypes and gene expression underlying seizure activity" by Ulrich Pfisterer and colleagues, describes neuronal transcriptome changes in cortical tissue from three patients with temporal lobe epilepsy and four non-epileptic controls. The authors use single-nucleus RNA sequencing in temporal cortex samples from a patient with epilepsy and compare to non-epileptic controls.

The basic approach used Smart-seq2 to create a reference dataset for the epileptic tissue. They then performed a joint analysis of healthy vs. epileptic tissue by 10XGenomics approach to identify neuronal subsets with significant epilepsy-associated changes in their transcriptomes.

By performing cortical layer-specific gene expression analyses, they were able to identify "hotspots" of hyperexcitability-related changes in upper layers and layer 5. These gene expression changes were associated with changes in upper and lower layers, including both excitatory glutamatergic neurons and GABAergic interneurons, most notably pyramidal neurons and VIP-expressing interneurons in layer 5.

The reference data set was based on temporal cortex from three patients with temporal lobe epilepsy that showed abnormalities in their MRIs in the temporal cortex. All cortical layers were included in the specimens and the neuronal nuclei were sorted on the basis of NeuN expression, as well as Sox2 labeling to enrich for GABAergic interneurons.

They then profiled the transcriptomes of over 1,000 neuronal nuclei using a modified version of SmartSeq2, that reduces biases of cDNA amplification and increases numbers of identified genes. The use of SmartSeq2 seems appropriate as a comparison of different single-cell sequencing methods has shown this approach to be the most sensitive method and most even coverage across all transcripts (Ziegenhain et al. 2017 <https://www.sciencedirect.com/science/article/pii/S1097276517300497>).

The major new findings of this study are epilepsy-associated transcriptional changes in layer 2 and layer 5 of the temporal cortex. This observation was supported by the finding that sets of co-expressed genes (gene modules) were significantly different in epileptic and non-epileptic cortex, based on applying a weighted gene co-expression network analysis across 20 cortical cell types. This led to the identification of 37 gene modules that were linked to the condition of epilepsy and all but 3 were down-regulated. Ten of these modules included co-expressed genes from the genetically-curated epilepsy gene list. They asked whether the altered gene modules were associated with synaptic functions (synapses, post-synaptic glutamate receptors, postsynaptic membrane potential), neuronal development, etc. The statistical analyses are appropriate and rigorous.

This approach suggested GABAergic interneuron dysfunction in epileptic circuits, particularly VIP-expressing interneurons. These findings are exciting, and noteworthy, as recent work has drawn attention to cortical VIP interneurons (which are not a single unified type, but rather multiple types, based on co-expression of neurotransmitters such as acetylcholine and serotonin receptors). VIP-INs receive long-range excitatory inputs, serotonergic and cholinergic afferents. Their roles in schizophrenia and epilepsy are of considerable interest, especially as they are strongly recruited by arousing events such as the onset of motor activity and they regulate cortical excitatory activity through inhibition of pyramidal neurons and other interneurons. Thus, this paper is likely to be of considerable interest to others working in related fields of neurology and neuroscience.

Despite the high quality and elegance of this study, several methodological problems reduce enthusiasm and should be addressed. In addition, the authors may wish to modify or dampen some of their interpretations for their findings, particularly their histological results and they need to more extensively refer to relevant work (as cited below).

Major concerns:

1) The authors appear to have based their reference data set on just three patients with temporal lobe epilepsy. We are given very little information about the patient's history and genetics. The authors need to provide a table of information about the patients, with information about the sex, and age of the patient at the time the tissue was obtained, the age of disease onset, the diagnosis, history of anti-convulsant drug treatments, the time from resection to freezing of the samples, and whether the patient was on medication at the time of biopsy, since these factors could alter the expression of epilepsy-related genes. The authors should also qualify their conclusions by clearly stating that their evidence is based on very limited sample sizes and note the heterogeneity in terms of disease onset, treatments, sex, and age.

2) The use of NeuN to identify neuronal nuclei and sort them by FACS could introduce certain biases in the dataset. NeuN staining identifies RBFOX3 (RNA binding fox-1 homolog 3) protein, an RNA binding protein that regulates alternative splicing of pre-mRNA. Prior studies in human and rodent cortical and hippocampal tissue from epileptic specimens linked reductions in the expression of NeuN to seizure foci and transcriptional changes. In particular in a study that examined a large number of specimens from patients with epilepsy (Dachet et al (Brain 2015 138:356) decreased NeuN immunoreactivity was found in layers and regions of the human cortex that were more likely to be high-spiking and show transcriptional changes indicative of epilepsy. Thus lower NeuN was demonstrated in hyperexcitable epileptic cortical tissue. Interestingly, these regions were also enriched in epilepsy biomarkers, and showed a concomitant increase in microglia, decreased MAP2+ neuropil, and increased vascularization, suggesting neuropathological changes caused by seizures. The authors of the previous study suggest that low NeuN microlesions, whatever their causes, correspond to areas that are highly epileptic. The irregular regions of low NeuN staining were not laminar specific and corresponded to areas of higher spiking activity, indicative of epileptic cortex (See Figure 4). They showed that the loss of NeuN was not due to cell death of the cortical neurons, but rather reduced expression of NeuN. Interestingly, these low-NeuN regions were adjacent to cortical layers that showed activation of Phospho-Creb and pERK - markers of higher neuronal activity.

a. These findings raise a methodological concern: due to the selective use of NeuN labeling to purify nuclei from the epileptic brain sample, has non-epileptic cortical tissue been preferentially selected? The authors of the present study need to consider how their dataset may be influenced by the use of NeuN to purify/enrich cortical neurons from the epilepsy patient. Does the data set over-represent less affected cells, since decreased NeuN in cortical neurons in multiple focal sites in the human temporal lobe from epilepsy patients corresponded to higher-spiking cortical areas that were associated with epileptic seizures?

b. The authors need to address the choice of their selection procedure and consider that it may have altered their findings, since they may have selected for the cells (high NeuN) that are not the most severely affected by epilepsy.

3) In Figure 6, the authors attempt to link pathological changes in the patient tissue to the findings in the reference dataset. The data presented and the interpretations are problematic.

a. The basis for stating that the cells have an abnormal orientation is unclear. How many cells were found that were atypical? What cortical areas, and layers? Given that this is one specimen, the authors should consider the possibility that focal cortical dysplasia, originating from a developmental defect in the cortical migration in this individual, was the underlying cause of epilepsy. In other words, causality cannot be inferred, given that it is unknown whether the individual had this abnormal region of cortex prior to the onset of epilepsy.

b. Interpretation of the changes in NeuN staining is a bit one-sided. It is impossible to infer that seizures caused these changes in NeuN, given the prior work in the literature establishing that NeuN expression is variable in cortex from epilepsy patients. Please provide a more nuanced discussion of the histological findings. In particular, causality is difficult to establish- are these

low-NeuN expressing neurons a cause of the seizures or result from seizure activity? For example, regions and individual cortical neurons with low NeuN expression may have distinctly different patterns of gene expression than the samples that were selected for in the present study.

c. How many patient samples and areas were analyzed?

d. Please quantify the abnormal orientations of neurons in a rigorous manner.

Minor issues:

4) Figure 6:

a. The scale bar length is not indicated

b. Panels d and d1 are out of focus

c. Panel e appears to be from the same tissue but is not indicated in the lower magnification views. It appears to be from a thinner or different kind of section (possible paraffin) and is possibly thinner and the image is considerably crisper than the other two.

d. Antibody used in the staining (NeuN) is not indicated in the figure caption.

e. Panel c has "d" indicated in two places - one in white font and one in black font

f. Panel showing neuronal depletion in "d" white font, doesn't appear to be in layer 5, so unclear what layer this represents. Need more definitive and reliable way to identify cortical layers (myelination stains and/or cytoarchitecture using Nissl stains, for example).

5) Overall, the writing is fine, but the authors need to check for punctuation and missing articles of speech.

6) Overall, the figures need to be more clearly labeled, small fonts are very difficult to read.

Reviewer #2 (Remarks to the Author):

General Summary

Evidence from rodent and human studies indicates that deficits in specific neuronal subtypes can lead to epilepsy. Presumably, genetic abnormalities within those neuronal subpopulations contributes to the proconvulsive phenotype. The authors of the current manuscript aim to provide a more resolved characterization of the genetic profiles of neuronal subtypes among healthy and epileptic humans. In their current study, the authors probe gene expression profiles at the single cell level. In this vein, the results are novel and important.

While generally descriptive and somewhat confirmatory, it is noteworthy that the single cell study was performed on humans and, therefore, provides an important and clearly clinically relevant resource for the epilepsy research community. Moreover, two independent measures were used to derive similar conclusions: i.e. Lamp5, Rorb, Grin3a, and VIP subtypes are abnormal in epileptic tissue. The conclusion that VIP neurons may represent a particularly susceptible node is interesting and consistent with recent reports in rodents. Finally, the authors provide information regarding the specific genes that underlie differences between normal and epileptic neuron subtypes.

Specific comments

1. Nuclei were sorted from microdissected epileptic cortical layers and transcriptome clusters were validated using layer-specific markers found in healthy cortices. Were individual cortical layers microdissected, thereby providing the confidence that neuronal subtypes located within specific layers of healthy tissue are found in the same layers of epileptic tissue? For example, can one unequivocally state that Lamp5-positive neurons from epileptic tissue are found in L2, as they are in healthy tissue? As tissue can reorganize in the epileptic brain (e.g. Fig 6), careful designation of layer specificity seems to be an important consideration. If one cannot confidently ascribe a neuron subtype to a layer, then perhaps the layer designations found in Figure 1 should be

removed.

2.Line 178: The explanation for missing neurons in epileptic tissue is not clear. Are the neurons missing, or are the neurons simply missing the marker? The discussion does not appear to address this curious finding.

3.Figure legends for supplemental figures do not provide much detail, making it somewhat difficult to understand what is shown.

4.Line 282/Fig 4: How are the genes in Figure 4b listed? By degree of modulation?

5.The claim that the histology validates the transcriptomic data is weak, as only general cell markers are used. The histological data are perhaps consistent with a few transcriptomic observations, but do not validate the entire approach.

6.Line 357: the authors are generally careful to note that any transcriptional abnormalities observed in epileptic tissue might not cause the disease, but rather result from the disease. However, starting on line 359 the authors begin describing their strategy for finding seizure-causing neuronal subtypes. Is this overstated, especially considering that the tissue in question was removed only to gain access to the likely seizure-causing tissue in the epileptogenic hippocampus? Presumably, it is the removal of the epileptogenic hippocampus that provided seizure freedom for patients, correct?

Minor

1.ABI presumably refers to Allen Brain Institute. The first instance of the acronym should include the full name.

2.What is PMI?

3.The manuscript is generally well written. However, definite articles (e.g. "the") are missing in several instances (e.g. Line 472). The authors may want to consider having a native-English writer proofread.

Reviewer #3 (Remarks to the Author):

The authors used the single-nucleus RNA sequencing method to depict a complex transcriptomic change in specific cell type of different layers of cortex underlying TLE.

Their work revealed epilepsy-related transcriptional changes across different subtypes, showed hotspots of hyperexcitability-related transcriptional changes in L2/5, with the conclusion that a coordinated dysregulation of gene expression in local circuits rather than individual subtypes. The topic is very interesting. The finding is consistent with the classic hypothesis "imbalance of excitability and inhibition" in epilepsy, but in a more a more comprehensive single-cell gene level. However, there are still some concerns see below:

(1) The present study is more likely to verify the transcriptional change of epilepsy-related genes that have been revealed before. The TRUE novel genes (not are reported to correlated with epilepsy before) is absence. If there is, it would be a nice novelty to the present work.

(2) Epilepsy is a highly heterogeneous disease with many causes. The current study includes only three TLE patients (1 FCD and 2 non-FCD), the sample size is too small.

(3) The characteristics of epilepsy patients should be more specific in detail, e.g. seizure frequency, seizure type, the time of sample collection after surgery, et al.. whether these factors would have impact on transcriptional identity.

(4) The immunohistochemistry in fig. 6 only revealed structural changes in cortex, which is very

weak to support for the research named "identification of epilepsy-associated neuronal subtypes and gene expression underlying epilepsy", unless the authors proved the changes of hotspots genes in specific neuronal subtypes mentioned before in immunohistochemistry.

(5) Is there any epilepsy-related transcriptional change in chloride transporters, e.g. NKCC1 or KCC2, which is critical for GABAergic inhibition?

Reviewer #4 (Remarks to the Author):

In this study, Pfisterer and colleagues have employed a single-cell transcriptomics approach to identify neuronal subtype specific changes in gene expression in the epileptic cortex. Specifically, the authors used both plate-based Smart-seq2 and droplet microfluidics-based 10x Genomics approaches to study a set of epileptic temporal cortex from temporal lobe epilepsy patient and non-epileptic control samples. Computational analyses revealed neuronal subtypes (including both excitatory and inhibitory neurons) that are associated with epilepsy-related transcriptional changes and previously identified disease phenotype related genes. Overall, the results from this study will be a useful resource for understanding the molecular and cellular mechanisms underlying epilepsy.

Specific comments:

1. Fig. 2: the number of nuclei analyzed in three epileptic samples by SMART-seq2 is much smaller than the number of healthy nuclei from the published ABI dataset. Importantly, some of neuronal subtypes cannot be identified in epileptic samples probably due to the relatively small number of nuclei analyzed (~1,000 nuclei). Thus, the comparison between epileptic and healthy single-nucleus RNA-seq may be biased, which make the interpretation of results reported in Fig. 2b-d very difficult.
2. Fig. 3: the authors should report the relative percentage of each neuronal subtypes in all three epileptic and four healthy samples as bar graphs for direct comparison. Also, it is unclear how the significance cutoff was chosen for Fig. 3c/d.
3. Fig. 4: Instead of showing many examples using violin plots, the authors should directly report the statistical significance and the number of differentially expressed genes between healthy and epileptic samples using unbiased analysis such as volcano plots for major neuronal subtypes.
4. The validation experiment is quite weak and it is unclear how the observed morphological changes are related to epilepsy-related genes that are enriched in L2/L5 excitatory neuronal subtypes. The authors should directly validate the expression of a set of candidate genes identified in healthy and epileptic cortex using the RNA FISH assay.
5. Several pioneering method development studies have established the droplet microfluidics-based massively parallel single-nucleus RNA-seq method and demonstrated the utility of this approach in analyzing frozen-archived brain samples at single-cell resolution (Habib et al, PMID: 28846088; Lake et al, PMID: 29227469; Hu et al, PMID: 29220646). This manuscript should cite these papers in the introduction section.
6. The summary statistics for all single-nucleus sequencing experiments isn't well documented by the authors. To better evaluate the data as a useful resource, the number of reads/UMIs/genes per sample (including epileptic and healthy samples from this study as well as the published work) should be reported for both SMART-seq2 and 10x Genomics platforms in a supplemental table.

We thank the reviewers for valuable comments and for generous support of our manuscript. To address all reviewers' concerns, we significantly expanded the number of analyzed patients and the revised version contains datasets for 17 patients (8 controls and 9 temporal lobe epilepsy). Furthermore, the joint dataset has >100,000 sequenced single neurons, thus being the largest dataset for human brain disorders published so far. Importantly, we validated our single nucleus RNAseq data by comprehensive single molecule fluorescent in situ hybridization (smFISH) analysis for multiple markers, confirming both cell type-specific and layer-specific changes in neuronal transcriptome due to epilepsy. Finally, we focus our manuscript on novel findings in epilepsy research - from novel molecules and protein families to novel circuits and neuronal networks.

To incorporate all new data, the revised manuscript had to be significantly expanded and re-written, and now includes 8 main figures, 17 supplementary figures and 19 supplementary tables.

We hope our changes address the issues raised by the reviewers. Please find below, our point-by-point reply.

Reviewers' comments:

Reviewer #1 (Remarks to the Author):

The manuscript submitted to Nature Communications, entitled "Identification of epilepsy-associated neuronal subtypes and gene expression underlying seizure activity" by Ulrich Pfisterer and colleagues, describes neuronal transcriptome changes in cortical tissue from three patients with temporal lobe epilepsy and four non-epileptic controls. The authors use single-nucleus RNA sequencing in temporal cortex samples from a patient with epilepsy and compare to non-epileptic controls.

The basic approach used Smart-seq2 to create a reference dataset for the epileptic tissue. They then performed a joint analysis of healthy vs. epileptic tissue by 10XGenomics approach to identify neuronal subsets with significant epilepsy-associated changes in their transcriptomes.

By performing cortical layer-specific gene expression analyses, they were able to identify "hotspots" of hyperexcitability-related changes in upper layers and layer 5. These gene expression changes were associated with changes in upper and lower layers, including both excitatory glutamatergic neurons and GABAergic interneurons, most notably pyramidal neurons and VIP-expressing interneurons in layer 5.

The reference data set was based on temporal cortex from three patients with temporal lobe epilepsy that showed abnormalities in their MRIs in the temporal cortex. All cortical layers were included in the specimens and the neuronal nuclei were sorted on the basis of NeuN expression, as well as Sox2 labeling to enrich for GABAergic interneurons.

They then profiled the transcriptomes of over 1,000 neuronal nuclei using a modified version of SmartSeq2, that reduces biases of cDNA amplification and increases numbers of identified genes. The use of SmartSeq2 seems appropriate as a comparison of different single-cell sequencing methods has shown this approach to be the most sensitive method and most even coverage across all transcripts (Ziegenhain et al. 2017 <https://www.sciencedirect.com/science/article/pii/S1097276517300497>).

The major new findings of this study are epilepsy-associated transcriptional changes in layer 2 and layer 5 of the temporal cortex. This observation was supported by the finding that sets of co-

expressed genes (gene modules) were significantly different in epileptic and non-epileptic cortex, based on applying a weighted gene co-expression network analysis across 20 cortical cell types. This led to the identification of 37 gene modules that were linked to the condition of epilepsy and all but 3 were down-regulated. Ten of these modules included co-expressed genes from the genetically-curated epilepsy gene list. They asked whether the altered gene modules were associated with synaptic functions (synapses, post-synaptic glutamate receptors, postsynaptic membrane potential), neuronal development, etc. The statistical analyses are appropriate and rigorous.

This approach suggested GABAergic interneuron dysfunction in epileptic circuits, particularly VIP-expressing interneurons. These findings are exciting, and noteworthy, as recent work has drawn attention to cortical VIP interneurons (which are not a single unified type, but rather multiple types, based on co-expression of neurotransmitters such as acetylcholine and serotonin receptors). VIP-INs receive long-range excitatory inputs, serotonergic and cholinergic afferents. Their roles in schizophrenia and epilepsy are of considerable interest, especially as they are strongly recruited by arousing events such as the onset of motor activity and they regulate cortical excitatory activity through inhibition of pyramidal neurons and other interneurons. Thus, this paper is likely to be of considerable interest to others working in related fields of neurology and neuroscience.

Despite the high quality and elegance of this study, several methodological problems reduce enthusiasm and should be addressed. In addition, the authors may wish to modify or dampen some of their interpretations for their findings, particularly their histological results and they need to more extensively refer to relevant work (as cited below).

Major

concerns:

1) The authors appear to have based their reference data set on just three patients with temporal lobe epilepsy. We are given very little information about the patient's history and genetics. The authors need to provide a table of information about the patients, with information about the sex, and age of the patient at the time the tissue was obtained, the age of disease onset, the diagnosis, history of anti-convulsant drug treatments, the time from resection to freezing of the samples, and whether the patient was on medication at the time of biopsy, since these factors could alter the expression of epilepsy-related genes. The authors should also qualify their conclusions by clearly stating that their evidence is based on very limited sample sizes and note the heterogeneity in terms of disease onset, treatments, sex, and age.

We agree with the reviewer about the limited number of samples in the previous version of the manuscript. In the revised manuscript, we have substantially increased the size of the dataset, now consisting of 9 temporal lobe epilepsy samples and 8 controls. All 9 temporal lobe epilepsy samples were derived from patients where epileptic activity has been confirmed in the temporal cortex by MRI. Furthermore, all patients except one had their epilepsy debut in their childhood. The size of the dataset is now comparable to the recently published single cell transcriptomic datasets for Alzheimer (Mathys et al. Nature 2019) and Multiple sclerosis (Schirmer et al. Nature 2019), and is even larger than Grubman et al. Nat Neurosci 2019 for Alzheimer. Additionally, we confirmed that such confounders as age and sex do not have a large impact on our cell type annotation. Nevertheless, we agree with the reviewer that to identify and account for changes that are related to sex, age and in particular medication, patient cohorts with limited variability for each confounder have to be studied. However, this is often very difficult (for some confounders – impossible) to control reliably nowadays due to limited cohorts that are available in tissue banks. Thus, we added a sentence in the Results section, where we state that much larger cohorts are necessary to account for major confounders.

We also apologize for providing too little information about the patients. This was due to waiting for approval from Data Protection Agency in Denmark for inclusion of patient data. We now obtained such approval and included extensive patient information in the Supplementary Table 1.

2) The use of NeuN to identify neuronal nuclei and sort them by FACS could introduce certain biases in the dataset. NeuN staining identifies RBFOX3 (RNA binding fox-1 homolog 3) protein, an RNA binding protein that regulates alternative splicing of pre-mRNA. Prior studies in human and rodent cortical and hippocampal tissue from epileptic specimens linked reductions in the expression of NeuN to seizure foci and transcriptional changes. In particular in a study that examined a large number of specimens from patients with epilepsy (Dachet et al (Brain 2015 138:356) decreased NeuN immunoreactivity was found in layers and regions of the human cortex that were more likely to be high-spiking and show transcriptional changes indicative of epilepsy. Thus lower NeuN was demonstrated in hyperexcitable epileptic cortical tissue. Interestingly, these regions were also enriched in epilepsy biomarkers, and showed a concomitant increase in microglia, decreased MAP2+ neuropil, and increased vascularization, suggesting neuropathological changes caused by seizures. The authors of the previous study suggest that low NeuN microlesions, whatever their causes, correspond to areas that are highly epileptic. The irregular regions of low NeuN staining were not laminar specific and corresponded to areas of higher spiking activity, indicative of epileptic cortex (See Figure 4). They showed that the loss of NeuN was not due to cell death of the cortical neurons, but rather reduced expression of NeuN. Interestingly, these low-NeuN regions were adjacent to cortical layers that showed activation of Phospho-Creb and pERK - markers of higher neuronal activity.

a. These findings raise a methodological concern: due to the selective use of NeuN labeling to purify nuclei from the epileptic brain sample, has non-epileptic cortical tissue been preferentially selected? The authors of the present study need to consider how their dataset may be influenced by the use of NeuN to purify/enrich cortical neurons from the epilepsy patient. Does the data set over-represent less affected cells, since decreased NeuN in cortical neurons in multiple focal sites in the human temporal lobe from epilepsy patients corresponded to higher-spiking cortical areas that were associated with epileptic seizures?

We thank the reviewer for noting that the neurons in the epileptic regions might have decreased expression of NeuN, which is certainly a concern for the study that use NeuN as a neuronal marker. In the revised version, we provide comprehensive evidence that our study is not biased towards high NeuN-expression cells and does consider all neurons in the epileptic samples. When selecting NeuN-positive nuclei during sorting, our approach was to expand the selection gates at Flow Sorter towards cells that were NeuN-negative, thus avoiding losing neuronal nuclei. This naturally resulted in inclusion of some glial-derived nuclei that were also sequenced later. However, glial nuclei could be easily removed from the analysis of single cell datasets based on marker expression. In the revised version, we provide flow cytometry images describing our nuclei selection protocol.

Moreover, to provide further evidence that we do not lose neurons in NeuN-negative flow cytometry gates, we performed single nucleus RNA sequencing for the NeuN-negative population and confirmed a negligible amount of neuronal nuclei in this population. The data is provided in the Supplementary Fig. 1.

b. The authors need to address the choice of their selection procedure and consider that it may have altered their findings, since they may have selected for the cells (high NeuN) that are not the most severely affected by epilepsy.

Please refer to our answer above.

3) In Figure 6, the authors attempt to link pathological changes in the patient tissue to the findings in the reference dataset. The data presented and the interpretations are problematic.

- a. The basis for stating that the cells have an abnormal orientation is unclear. How many cells were found that were atypical? What cortical areas, and layers? Given that this is one specimen, the authors should consider the possibility that focal cortical dysplasia, originating from a developmental defect in the cortical migration in this individual, was the underlying cause of epilepsy. In other words, causality cannot be inferred, given that it is unknown whether the individual had this abnormal region of cortex prior to the onset of epilepsy.

Since the analysis in the Figure 6 of the original submission received high critique from most reviewers, we decided to remove this part of work from the manuscript. Instead, in the revised version, we provide comprehensive validation of our single nucleus transcriptomics results with single molecule fluorescent in situ hybridization (smFISH) for multiple markers. Importantly, smFISH confirms vast changes in the neuronal transcriptome in the epileptic cortices. The new data is included in the Figures 5, 6 and 7 in the revised version.

- b. Interpretation of the changes in NeuN staining is a bit one-sided. It is impossible to infer that seizures caused these changes in NeuN, given the prior work in the literature establishing that NeuN expression is variable in cortex from epilepsy patients. Please provide a more nuanced discussion of the histological findings. In particular, causality is difficult to establish– are these low-NeuN expressing neurons a cause of the seizures or result from seizure activity? For example, regions and individual cortical neurons with low NeuN expression may have distinctly different patterns of gene expression than the samples that were selected for in the present study.
- c. How many patient samples and areas were analyzed?
- d. Please quantify the abnormal orientations of neurons in a rigorous manner.

Please refer to our answer to point 3a above, where we describe the reasons for removing of the histological data.

Minor issues:

4) Figure 6:

- a. The scale bar length is not indicated
- b. Panels d and d1 are out of focus
- c. Panel e appears to be from the same tissue but is not indicated in the lower magnification views. It appears to be from a thinner or different kind of section (possible paraffin) and is possibly thinner and the image is considerably crisper than the other two.
- d. Antibody used in the staining (NeuN) is not indicated in the figure caption.
- e. Panel c has "d" indicated in two places - one in white font and one in black font
- f. Panel showing neuronal depletion in "d" white font, doesn't appear to be in layer 5, so unclear what layer this represents. Need more definitive and reliable way to identify cortical layers (myelination stains and/or cytoarchitecture using Nissl stains, for example).

Please refer to our answer to point 3a above, where we describe the reasons for removing the histological data.

5) Overall, the writing is fine, but the authors need to check for punctuation and missing articles of speech.

We thank the reviewer for noting some issues with punctuation and grammar. The manuscript has been proofread by an English native speaker.

6) Overall, the figures need to be more clearly labeled, small fonts are very difficult to read.

We thank the reviewer for helping us to improve our manuscript. We improved figure legibility and increased font size where possible.

Reviewer #2 (Remarks to the Author):

General Summary

Evidence from rodent and human studies indicates that deficits in specific neuronal subtypes can lead to epilepsy. Presumably, genetic abnormalities within those neuronal subpopulations contribute to the proconvulsive phenotype. The authors of the current manuscript aim to provide a more resolved characterization of the genetic profiles of neuronal subtypes among healthy and epileptic humans. In their current study, the authors probe gene expression profiles at the single cell level. In this vein, the results are novel and important.

While generally descriptive and somewhat confirmatory, it is noteworthy that the single cell study was performed on humans and, therefore, provides an important and clearly clinically relevant resource for the epilepsy research community. Moreover, two independent measures were used to derive similar conclusions: i.e. Lamp5, Rorb, Grin3a, and VIP subtypes are abnormal in epileptic tissue. The conclusion that VIP neurons may represent a particularly susceptible node is interesting and consistent with recent reports in rodents. Finally, the authors provide information regarding the specific genes that underlie differences between normal and epileptic neuron subtypes.

We thank the reviewer for positive comments highlighting that our study is “clearly clinically relevant resource for the epilepsy research community”. In the revised version, we expanded our study to >100,000 neuronal nuclei and added a significant number of patient samples. We further focus our study on novel findings related to epilepsy, from novel genes and protein families to novel circuits. We validate our findings with single molecule fluorescent in situ hybridization (smFISH) for multiple markers, which confirms the changes in the transcriptome that were found by single nucleus transcriptomics.

Specific comments

1. Nuclei were sorted from microdissected epileptic cortical layers and transcriptome clusters were validated using layer-specific markers found in healthy cortices. Were individual cortical layers microdissected, thereby providing the confidence that neuronal subtypes located within specific layers of healthy tissue are found in the same layers of epileptic tissue? For example, can one unequivocally state that Lamp5-positive neurons from epileptic tissue are found in L2, as they are in healthy tissue? As tissue can reorganize in the epileptic brain (e.g. Fig 6), careful designation of layer specificity seems to be an important consideration. If one cannot confidently ascribe a neuron subtype to a layer, then perhaps the layer designations found in Figure 1 should be removed.

In the revised version, we established all layer-wise information based on the healthy/non-epileptic cortex and then related this layer-wise distribution to the data derived from epileptic cortex. For each subtype of neurons in our study, there was a good representation of neuronal nuclei coming from both healthy and epileptic cortices. Thus, we believe that the layer-wise designation of principal cell subtypes in our dataset is valid. Furthermore, as another validation of layer-wise annotation for our dataset, we related our subtypes to a previously published atlas of the healthy

human temporal cortex generated by the Allen Brain Institute and found a very good match of the layer designations between our annotation and theirs. Additionally, since the Allen Brain Institute data also assessed the layer position of GABAergic interneurons, such integration of our and the Allen Brain Institute's dataset allowed us to assign layer-wise positions to GABAergic interneurons in our annotation.

Please note that we qualified our statement regarding the layer designations by acknowledging that the layer positions for principal neurons are predictive.

2.Line 178: The explanation for missing neurons in epileptic tissue is not clear. Are the neurons missing, or are the neurons simply missing the marker? The discussion does not appear to address this curious finding.

In the revised version, we generated a much larger dataset for both non-epileptic and epileptic cortices, and thus we now removed comparison to Allen Brain Institute data and rather compared non-epileptic and epileptic cortices generated by the same single nucleus RNAseq protocol in the same lab. When we align non-epileptic and epileptic single nucleus transcriptomic data, the clustering of subtypes is based on the whole nuclear transcriptome rather than specific markers. If non-epileptic/epileptic nuclei are clustered together, these generally have a similar transcriptome. Thus, although specific markers are used for the annotation, the lack of a marker will not change the clustering of a nucleus to a certain cluster if the majority of the other genes are expressed in a subtype-specific manner.

The text for comparative analysis of absolute and relative neuronal number for each subtype across conditions was re-written and expanded in the revised version (now in Fig. 2b,c).

3.Figure legends for supplemental figures do not provide much detail, making it somewhat difficult to understand what is shown.

We thank the reviewer for pointing it out and in the revised version we expanded figure legends of both main and supplementary figures.

4.Line 282/Fig 4: How are the genes in Figure 4b listed? By degree of modulation?

In the revised version, we provide much more comprehensive analysis of pathways and GO terms and thus changed the outline of the figures. The analysis is now in Fig. 3, 4 and Supplementary Fig. 10-14

5.The claim that the histology validates the transcriptomic data is weak, as only general cell markers are used. The histological data are perhaps consistent with a few transcriptomic observations, but do not validate the entire approach.

We agree with the reviewer and in the revised version we provide comprehensive validation of transcriptomic changes by single molecule fluorescent in situ hybridization (smFISH) for multiple markers, which confirms the changes in the transcriptome that were found by single nucleus transcriptomics. The data is now in Figures 5, 6 and 7.

6.Line 357: the authors are generally careful to note that any transcriptional abnormalities observed in epileptic tissue might not cause the disease, but rather result from the disease. However, starting on line 359 the authors begin describing their strategy for finding seizure-causing neuronal subtypes. Is this overstated, especially considering that the tissue in question was removed only to gain access to the likely seizure-causing tissue in the epileptogenic hippocampus? Presumably, it is the removal of the epileptogenic hippocampus that provided seizure freedom for patients, correct?

We agree with the reviewer that in the temporal lobe epilepsy the primary role for epileptogenesis is usually assigned to the hippocampus. Furthermore, as the reviewer correctly writes, often the cortex is removed only to reach the focus in the hippocampus. However, when selecting the samples to analyse in our study, we focused only on those where the cortex also showed epileptic activity, which is indeed a minor part from the total number of TLE cases. Thus, all 9 temporal lobe epilepsy samples in the revised version were derived from patients where epileptic activity has been confirmed in the temporal cortex by MRI. We apologize that such important detail was not clear in the original version of the manuscript and provide clear description of how samples were selected in the first paragraph of the Results section.

Minor

1.ABI presumably refers to Allen Brain Institute. The first instance of the acronym should include the full name.

Thank you for noticing this. In the revised version, we use the abbreviation of Allen Brain Institute only in the Methods part when describing formulas and we included the acronym only after mentioning the full name.

2.What is PMI?

This is “postmortem interval” and we included the full name before we use the abbreviation in the Supplementary Table 1.

3.The manuscript is generally well written. However, definite articles (e.g. “the”) are missing in several instances (e.g. Line 472). The authors may want to consider having a native-English writer proofread.

We thank the reviewer for the suggestion and the revised version has been read through by a native English speaker.

Reviewer #3 (Remarks to the Author):

The authors used the single-nucleus RNA sequencing method to depict a complex transcriptomic change in specific cell type of different layers of cortex underlying TLE. Their work revealed epilepsy-related transcriptional changes across different subtypes, showed hotspots of hyperexcitability-related transcriptional changes in L2/5, with the conclusion that a coordinated dysregulation of gene expression in local circuits rather than individual subtypes. The topic is very interesting. The finding is consistent with the classic hypothesis “imbalance of excitability and inhibition” in epilepsy, but in a more a more comprehensive single-cell gene level. However, there are still some concerns see below:

(1) The present study is more likely to verify the transcriptional change of epilepsy-related genes that have been revealed before. The TRUE novel genes (not are reported to correlated with epilepsy before) is absence. If there is, it would be a nice novelty to the present work.

We apologize that we did not put enough emphasis on true novel epilepsy-related genes identified in our work. In fact, there are hundreds of novel genes and a number of pathways. Several of these we validated by single molecule fluorescent in situ hybridization (smFISH), including CKAMP44 (gene name Shisa9) and Grin3a. One of the most striking findings was increase in

expression of multiple genes coding for AMPA receptor auxiliary subunits and none of these genes we identified have been reported to be modulated in epilepsy before. In the revised version, we put a particular emphasis on novel epilepsy-related genes uncovered by our study. Overall, approx. 6,900 and 13,700 genes changed expression in at least one subtype of GABAergic interneurons or principal neurons, respectively (Supplementary Table 6), which is much larger than previously thought based on bulk RNA sequencing.

(2) Epilepsy is a highly heterogeneous disease with many causes. The current study includes only three TLE patients (1 FCD and 2 non-FCD), the sample size is too small.

We agree with the reviewer that epilepsy is a highly heterogeneous disease. Thus, in our study we tried to limit patient variability by studying only temporal lobe epilepsy patients and by including only those samples where epileptic activity has been confirmed in the temporal cortex by MRI. Furthermore, all patients except one had their epilepsy debut in their childhood.

We also agree that the original sample size was rather small. Therefore, in the revised version, we performed additional experiments and significantly expanded the analysis by including additional samples - the total dataset in the revised version contains 17 samples (8 controls and 9 temporal lobe epilepsy) and >100,000 neuronal nuclei.

Thus, we believe that our dataset in the revised version is state-of-the-art, being similar to recently published single cell transcriptomic datasets for Alzheimer (Mathys et al. Nature 2019) and Multiple sclerosis (Schirmer et al. Nature 2019), and much larger than Grubman et al. Nat Neurosci 2019 for Alzheimer.

(3) The characteristics of epilepsy patients should be more specific in detail, e.g. seizure frequency, seizure type, the time of sample collection after surgery, et al.. whether these factors would have impact on transcriptional identity.

We apologize that we could not provide too detailed patient information, which was due to privacy issues that are related to new GDPR (General Data Protection Regulation) rules in the European Union (even though we have full consent signed by each patient). Furthermore, we had to receive such permission for patient data publication from Danish National Data Protection Agency, which has additional rules that are applied only for Denmark. Before submitting the revised version, we did obtain permission to include some patient information e.g. history of anti-convulsant drugs and medication at the time of surgery, which allowed us to expand significantly patient information in the Supplementary Table 1. Some of the patient details that were requested by the reviewers we are not yet allowed to include to be published open for general public. We are working on it and should be able to include the rest of requested patient details in the final version of the manuscript before publication, given that we get permission from National Data Protection Agency.

(4) The immunohistochemistry in fig. 6 only revealed structural changes in cortex, which is very weak to support for the research named "identification of epilepsy-associated neuronal subtypes and gene expression underlying epilepsy", unless the authors proved the changes of hotspot genes in specific neuronal subtypes mentioned before in immunohistochemistry.

Since the analysis in Figure 6 of the original submission received high critique from most reviewers, we decided to remove this part of work from the manuscript. Instead, in the revised version, we provide comprehensive validation of our single nucleus transcriptomics results with single molecule fluorescent in situ hybridization (smFISH) for multiple markers. Importantly, smFISH confirms vast changes in neuronal transcriptome in the epileptic cortex. The new data are included in the Figures 5, 6 and 7 in the revised version.

(5) Is there any epilepsy-related transcriptional change in chloride transporters, e.g. NKCC1 or KCC2, which is critical for GABAergic inhibition?

Indeed, there were changes in expression of genes coding for proteins involved in chloride transport, both at single gene level as well as pathways/GO terms. The information about dysregulation of chloride transport genes is included in Supplementary Tables 6, 10 and 13.

Reviewer #4 (Remarks to the Author):

In this study, Pfisterer and colleagues have employed a single-cell transcriptomics approach to identify neuronal subtype specific changes in gene expression in the epileptic cortex. Specifically, the authors used both plate-based Smart-seq2 and droplet microfluidics-based 10x Genomics approaches to study a set of epileptic temporal cortex from temporal lobe epilepsy patient and non-epileptic control samples. Computational analyses revealed neuronal subtypes (including both excitatory and inhibitory neurons) that are associated with epilepsy-related transcriptional changes and previously identified disease phenotype related genes. Overall, the results from this study will be a useful resource for understanding the molecular and cellular mechanisms underlying epilepsy.

Specific comments:

1. Fig. 2: the number of nuclei analyzed in three epileptic samples by SMART-seq2 is much smaller than the number of healthy nuclei from the published ABI dataset. Importantly, some of neuronal subtypes cannot be identified in epileptic samples probably due to the relatively small number of nuclei analyzed (~1,000 nuclei). Thus, the comparison between epileptic and healthy single-nucleus RNA-seq may be biased, which make the interpretation of results reported in Fig. 2b-d very difficult.

We thank the reviewer for pointing to this issue. In the revised version, we generated a much larger dataset for both non-epileptic and epileptic cortices, and thus we now removed the comparison to the Allen Brain Institute data and rather compared non-epileptic and epileptic cortices generated by the same single nucleus RNAseq protocol in the same lab.

We also agree that the original sample size was rather small. Therefore, in the revised version, we performed additional experiments and significantly expanded the analysis by including many additional samples - the total dataset in the revised version is 17 samples (8 controls and 9 temporal lobe epilepsy) and >100,000 neuronal nuclei sequenced.

2. Fig. 3: the authors should report the relative percentage of each neuronal subtypes in all three epileptic and four healthy samples as bar graphs for direct comparison. Also, it is unclear how the significance cutoff was chosen for Fig. 3c/d.

Since in the revised version we analyzed many more samples, it might be too busy to include all samples in one graph and thus we report the percentage for each neuronal subtype in Supplementary Table 4. Furthermore, representation of each subtype and each sample is comprehensively described in Supplementary Fig. 2d-f.

The old Fig. 3c/d has been removed and similar type of analyses are now shown in Fig. 2d-f. We did not make any significance cut off in Fig. 2d-f, since any cut off would be arbitrary. Thus, we focus of those subtypes that have higher and lower values relative to each other. We also added a line in Fig. 2d that indicates average similarity score for all subtypes together, and in Fig. 2e,f - odds ratio equal to 1, which corresponds to "no difference observed".

3. Fig. 4: Instead of showing many examples using violin plots, the authors should directly report the statistical significance and the number of differentially expressed genes between healthy and epileptic samples using unbiased analysis such as volcano plots for major neuronal subtypes.

According to the reviewer suggestion, we show violin plots for differentially expressed genes in Supplementary Figures 9 and 10.

4. The validation experiment is quite weak and it is unclear how the observed morphological changes are related to epilepsy-related genes that are enriched in L2/L5 excitatory neuronal subtypes. The authors should directly validate the expression of a set of candidate genes identified in healthy and epileptic cortex using the RNA FISH assay.

We thank the reviewer for emphasizing the importance of validating single cell transcriptomics data by FISH. In the revised version, we performed comprehensive validation of our single nucleus transcriptomics results with single molecule fluorescent in situ hybridization (smFISH) for multiple markers. Importantly, smFISH confirms vast changes in neuronal transcriptome in the epileptic cortex. The new data is included in the Figures 5, 6 and 7 in the revised version.

5. Several pioneering method development studies have established the droplet microfluidics-based massively parallel single-nucleus RNA-seq method and demonstrated the utility of this approach in analyzing frozen-archived brain samples at single-cell resolution (Habib et al, PMID: 28846088; Lake et al, PMID: 29227469; Hu et al, PMID: 29220646). This manuscript should cite these papers in the introduction section.

We cite these highly relevant papers in the Introduction section.

6. The summary statistics for all single-nucleus sequencing experiments isn't well documented by the authors. To better evaluate the data as a useful resource, the number of reads/UMIs/genes per sample (including epileptic and healthy samples from this study as well as the published work) should be reported for both SMART-seq2 and 10x Genomics platforms in a supplemental table.

We apologize for omitting this important information from the original submission and we included the data in the Supplementary Table 2.

REVIEWERS' COMMENTS:

Reviewer #1 (Remarks to the Author):

MS212297_1 Identification of epilepsy-associated neuronal subtypes and gene expression underlying epileptogenesis

The major claims of the paper are that temporal cortex of epilepsy patients, compared to non-epileptic subjects, shows large changes in the transcriptomes with distinct types of neurons, including principal cells and interneurons. The authors find that epilepsy-related dysregulation of the transcriptome occurs in neuronal subtypes in a modular manner. The cells with the largest transcriptomic changes are within the same neural circuits, suggesting that they linked a coordinated dysregulation of gene expression within local circuits to epileptogenesis. One large difference occurred with glutamate signaling, particularly glutamate receptor genes, in addition to a strong up-regulation of AMPA receptor auxiliary subunits. The transcriptomics data were validated by single molecule Fluorescent In Situ Hybridization (FISH) analyses in patient tissue.

Single-cell transcriptomics is an important and rapidly developing technology that allows studies of how a pathological condition, such as epilepsy, can alter cellular composition and gene expression within specific brain regions. Remarkably, this study showed that epilepsy is characterized by the dysregulation of thousands of genes. Many of the dysregulated genes identified in this study have not been previously reported to be altered in epilepsy. Overall, these findings are novel, and these data will likely greatly interest others in the epilepsy community and the wider field.

This is a revision of an earlier manuscript that has been considerably strengthened by the addition of more patient samples. In this revised manuscript, samples from 19 epileptic and non-epileptic subjects (9 epileptic and 8 non-epileptics; 2 additional control samples were excluded based on low quality sequencing data) were micro dissected and analyzed using droplet-based 10X Genomics chemistry. Nearly 105,000 nuclei were sequenced and of these 81,586 passed stringent quality control and were found to be distinct neuronal subtypes. Using Conos to integrate the datasets and annotate them using layer-specific markers for different cell types, they found that 1 epilepsy patient and 2 non-epileptic samples showed bias due to absence of quite a few of the neuronal subtypes, so these samples were also excluded. The remaining samples did not show batch-specific effects and were well integrated. This final group of samples included 8 epileptic and 6 non-epileptic samples.

The single-nucleus transcriptomes analyzed by Conos were annotated, based on a hierarchical strategy that separated neurons into principal vs. GABAergic neurons, and these were further categorized into layer-specific groups (Cux2, Rorb, Themis, Fezf2) or in the case of GABAergic interneurons, into subsets expressing parvalbumin, somatostatin, vasoactive intestinal polypeptide, or Id2.

One of the key findings was a distinct shift in the transcriptomes of principal neuron types and a reduction in the number of L2/3 subtypes in epileptic tissue. The GABAergic interneuron populations were also altered, particularly striking was a loss of subtypes of Parvalbumin_ interneurons. They also used a gene-expression similarity scores based on the Pearson correlation of expression within and between conditions. Low similarity value showed a large difference between epileptic and non-epileptic neuronal subtypes. This analysis suggested that some subtypes in the epileptic cortex had low similarity with their counterparts in control samples, suggesting epilepsy-specific effects on the transcriptomes of particular cell types within certain cortical layers. For example. this effect was most striking for subtypes of principal neurons in Layer 5/6 and Layer 2/3. Likewise, when comparing GABAergic cell types, some subtypes were more affected than others, suggesting that epilepsy may disproportionately affect some types of interneurons more than others. While not a new idea, it is nice to see transcriptomic data that

supports previously-reported anatomical findings in epileptic tissue.

When they compared differentially expressed (DE) genes between TLE and control cortex samples, they also found some new and surprising results. They compared these DE genes to several gene lists curated based on genetic studies in human epilepsy patients and another from the largest epilepsy genome-wide-association study (GWAS) currently. This enrichment analysis confirmed the prevalence of epilepsy associated DE genes. Overall, when they calculated the enrichment of DE genes for Gene Ontology terms in each of the identified subtypes, they found that some neuronal subtypes exhibited large transcriptomic changes in epilepsy (>100 GO terms) whereas others had only a few. Clustering of the GO terms showed aberrant changes in the expression of genes involved in neuronal circuit re-organization and neurotransmission, including regulation of membrane potential, glutamatergic neurotransmission, synapse organization, and cell-cell adhesion DE genes, among others. The gene modules associated with epilepsy showed changes at the level of synapses and ion channels, including AMPA receptor auxiliary subunits, glutamate receptor subunits, voltage gated sodium channels that are likely involved in hyperexcitation in epileptic brain circuits.

One of the most interesting analyses performed in this study was to search for GO terms that would likely contribute to hyperexcitability of principal neurons or hypo inhibition of GABAergic interneurons. One novel finding they report is the identification of genes with large changes in expression that were not previously reported for epilepsy, including AMPA auxiliary subunit *CKAMP44*, a recently discovered member of this family that increases currents for AMPA receptors. This gene exhibited increased expression across all cortical layers in epileptic samples. Additional AMPA receptor auxiliary subunits were also upregulated in principal neurons and in GABAergic interneurons.

Overall, this manuscript provides new information regarding single cell transcriptome changes in epileptic tissue. This study provides a wealth of new information that will be useful to others working in the field.

Janice R. Naegele

Reviewer #2 (Remarks to the Author):

The authors have done great job addressing my comments. I remain supportive of the study, and I believe the revised manuscript is much stronger.

One major question remains: are the observed differences between healthy and epileptic tissue driving seizures, or do they result from seizures? Answering this question goes far beyond the scope of this study. Indeed, it's a decades-old question. The authors do a reasonable job at acknowledging this unanswered question in the Results section. Perhaps the authors could expand on this topic in the discussion (currently, there's only one line devoted to this question: Line 603).

If this major question remains unanswered, then is the authors' study still impactful? I think so. The authors have done a careful, in-depth analysis at the single cell level in human tissue. I feel that this study will provide an important transcriptional map of temporal lobe epilepsy, and will therefore be of significant value for the epilepsy research community.

The authors may want to discuss their findings in the context of a relatively recent review by Staley (<https://www.ncbi.nlm.nih.gov/pmc/articles/PMC4409128/>). This review seems particularly relevant for the authors' identification of novel, epilepsy-related genes (Line 511).

Finally, the authors have provided an explanation of how changes in glutamate receptors may

contribute to seizures. The authors have also identified changes in genes associated with action potentials (Fig 4b), but do not provide much discussion of what these changes might mean. For completeness, the authors may want to devote some discussion to action potential-related genes.

In sum, I think the authors provide a thorough, interesting study on transcriptional changes in epileptic tissue. I remain quite supportive.

Reviewer #3 (Remarks to the Author):

The revised manuscript has been improve a lot.
I have no additional concern.

Reviewer #4 (Remarks to the Author):

In the revised manuscript, the authors have satisfactorily addressed my previous concerns.

REVIEWERS' COMMENTS:

Reviewer #1 (Remarks to the Author):

MS212297_1 Identification of epilepsy-associated neuronal subtypes and gene expression underlying epileptogenesis

The major claims of the paper are that temporal cortex of epilepsy patients, compared to non-epileptic subjects, shows large changes in the transcriptomes with distinct types of neurons, including principal cells and interneurons. The authors find that epilepsy-related dysregulation of the transcriptome occurs in neuronal subtypes in a modular manner. The cells with the largest transcriptomic changes are within the same neural circuits, suggesting that they linked a coordinated dysregulation of gene expression within local circuits to epileptogenesis. One large difference occurred with glutamate signaling, particularly glutamate receptor genes, in addition to a strong up-regulation of AMPA receptor auxiliary subunits. The transcriptomics data were validated by single molecule Fluorescent In Situ Hybridization (FISH) analyses in patient tissue.

Single-cell transcriptomics is an important and rapidly developing technology that allows studies of how a pathological condition, such as epilepsy, can alter cellular composition and gene expression within specific brain regions. Remarkably, this study showed that epilepsy is characterized by the dysregulation of thousands of genes. Many of the dysregulated genes identified in this study have not been previously reported to be altered in epilepsy. Overall, these findings are novel, and these data will likely greatly interest others in the epilepsy community and the wider field.

This is a revision of an earlier manuscript that has been considerably strengthened by the addition of more patient samples. In this revised manuscript, samples from 19 epileptic and non-epileptic subjects (9 epileptic and 8 non-epileptics; 2 additional control samples were excluded based on low quality sequencing data) were micro dissected and analyzed using droplet-based 10X Genomics chemistry. Nearly 105,000 nuclei were sequenced and of these 81,586 passed stringent quality control and were found to be distinct neuronal subtypes. Using Conos to integrate the datasets and annotate them using layer-specific markers for different cell types, they found that 1 epilepsy patient and 2 non-epileptic samples showed bias due to absence of quite a few of the neuronal subtypes, so these samples were also excluded. The remaining samples did not show batch-specific effects and were well integrated. This final group of samples included 8 epileptic and 6 non-epileptic samples.

The single-nucleus transcriptomes analyzed by Conos were annotated, based on a hierarchical strategy that separated neurons into principal vs. GABAergic neurons, and these were further categorized into layer-specific groups (Cux2, Rorb, Themis, Fezf2) or in the case of GABAergic interneurons, into subsets expressing parvalbumin, somatostatin, vasoactive intestinal polypeptide, or Id2.

One of the key findings was a distinct shift in the transcriptomes of principal neuron types and a reduction in the number of L2/3 subtypes in epileptic tissue. The GABAergic interneuron populations were also altered, particularly striking was a loss of subtypes of Parvalbumin_ interneurons. They also used a gene-expression similarity scores based on the Pearson correlation of expression within and between conditions. Low similarity value showed a large difference between epileptic and non-epileptic neuronal subtypes. This analysis suggested that some subtypes in the epileptic cortex had low similarity with their counterparts in control samples, suggesting epilepsy-specific effects on the transcriptomes of particular cell types within certain cortical layers. For example. this effect was most striking for subtypes of principal neurons in Layer 5/6 and Layer 2/3. Likewise, when comparing GABAergic cell types, some subtypes were more affected than

others, suggesting that epilepsy may disproportionately affect some types of interneurons more than others. While not a new idea, it is nice to see transcriptomic data that supports previously-reported anatomical findings in epileptic tissue.

When they compared differentially expressed (DE) genes between TLE and control cortex samples, they also found some new and surprising results. They compared these DE genes to several gene lists curated based on genetic studies in human epilepsy patients and another from the largest epilepsy genome-wide-association study (GWAS) currently. This enrichment analysis confirmed the prevalence of epilepsy associated DE genes. Overall, when they calculated the enrichment of DE genes for Gene Ontology terms in each of the identified subtypes, they found that some neuronal subtypes exhibited large transcriptomic changes in epilepsy (>100 GO terms) whereas others had only a few. Clustering of the GO terms showed aberrant changes in the expression of genes involved in neuronal circuit re-organization and neurotransmission, including regulation of membrane potential, glutamatergic neurotransmission, synapse organization, and cell-cell adhesion DE genes, among others. The gene modules associated with epilepsy showed changes at the level of synapses and ion channels, including AMPA receptor auxiliary subunits, glutamate receptor subunits, voltage gated sodium channels that are likely involved in hyperexcitation in epileptic brain circuits.

One of the most interesting analyses performed in this study was to search for GO terms that would likely contribute to hyperexcitability of principal neurons or hypo inhibition of GABAergic interneurons. One novel finding they report is the identification of genes with large changes in expression that were not previously reported for epilepsy, including AMPA auxiliary subunit CKAMP44, a recently discovered member of this family that increases currents for AMPA receptors. This gene exhibited increased expression across all cortical layers in epileptic samples. Additional AMPA receptor auxiliary subunits were also upregulated in principal neurons and in GABAergic interneurons.

Overall, this manuscript provides new information regarding single cell transcriptome changes in epileptic tissue. This study provides a wealth of new information that will be useful to others working in the field.

Janice R. Naegele

We thank the reviewer for positive and encouraging comments.

Reviewer #2 (Remarks to the Author):

The authors have done great job addressing my comments. I remain supportive of the study, and I believe the revised manuscript is much stronger.

One major question remains: are the observed differences between healthy and epileptic tissue driving seizures, or do they result from seizures? Answering this question goes far beyond the scope of this study. Indeed, it's a decades-old question. The authors do a reasonable job at acknowledging this unanswered question in the Results section. Perhaps the authors could expand on this topic in the discussion (currently, there's only one line devoted to this question: Line 603).

If this major question remains unanswered, then is the authors' study still impactful? I think so. The authors have done a careful, in-depth analysis at the single cell level in human tissue. I feel that this study will provide an important transcriptional map of temporal lobe epilepsy, and will therefore be of significant value for the epilepsy research community.

The authors may want to discuss their findings in the context of a relatively recent review by Staley (<https://www.ncbi.nlm.nih.gov/pmc/articles/PMC4409128/>). This review seems particularly relevant for the authors' identification of novel, epilepsy-related genes (Line 511).

Finally, the authors have provided an explanation of how changes in glutamate receptors may contribute to seizures. The authors have also identified changes in genes associated with action potentials (Fig 4b), but do not provide much discussion of what these changes might mean. For completeness, the authors may want to devote some discussion to action potential-related genes.

In sum, I think the authors provide a thorough, interesting study on transcriptional changes in epileptic tissue. I remain quite supportive.

We thank the reviewer for generous comments and we changed the discussion according to reviewer's suggestions.

Reviewer #3 (Remarks to the Author):

*The revised manuscript has been improve a lot.
I have no additional concern.*

Reviewer #4 (Remarks to the Author):

In the revised manuscript, the authors have satisfactorily addressed my previous concerns.

We thank both reviewers for their positive comments.

REVIEWERS' COMMENTS

Reviewer #5 (Remarks to the Author):

This is a revision of the revised version of the manuscript "Identification of epilepsy-associated neuronal sub types and gene expression underlying epileptogenesis". The authors have included a significant number of samples, epileptic and non-epileptic (19 samples) and have performed a series of robust computational analyses. The raw data can be accessed through repositories for sensitive data as EGA and the code used in the study can be reuse through their github link.

The study combines integration of different single cell technologies, showing the strength of combining different sources of data. The methods and approaches show a very robust computational analysis. This reviewer thanks and liked the detailed methods section and considers this study of high value in the transcriptional characterization of epilepsy.

comments:

The authors used Scrublet for doublet removal. 10X genomics data is known for the high rates of doublets and more specifically in their v3. Based on experience, I would suggest the authors to show the UMAP embeddings with the doublets and to provide the score used for the doublet removal. Many of the called doublets could still be biological doublets, with combinations of non canonical marker genes. Do the authors perform any double check on this point? It would be helpful to know the proportion of doublets per sample.

On data availability, the authors provide very helpful raw data and the code used. I would suggest to also include annotated matrices with the identified sub types in order to help to the reproducibility and the use in other studies of their findings. Other studies included interactive visualizations or directly the cell IDs and cell types annotations.

REVIEWERS' COMMENTS:

Reviewer #5 (Remarks to the Author):

This is a revision of the revised version of the manuscript "Identification of epilepsy-associated neuronal sub types and gene expression underlying epileptogenesis". The authors have included a significant number of samples, epileptic and non-epileptic (19 samples) and have performed a series of robust computational analyses. The raw data can be accessed through repositories for sensitive data as EGA and the code used in the study can be reuse through their github link.

The study combines integration of different single cell technologies, showing the strength of combining different sources of data. The methods and approaches show a very robust computational analysis. This reviewer thanks and liked the detailed methods section and considers this study of high value in the transcriptional characterization of epilepsy.

We thank the reviewer for their positive comments.

comments:

The authors used Scrublet for doublet removal. 10X genomics data is known for the high rates of doublets and more specifically in their v3. Based on experience, I would suggest the authors to show the UMAP embeddings with the doublets and to provide the score used for the doublet removal. Many of the called doublets could still be biological doublets, with combinations of non canonical marker genes. Do the authors perform any double check on this point? It would be helpful to know the proportion of doublets per sample.

According to reviewer suggestion, we added UMAP plots with doublets highlighted, provided the score and proportions of doublets per sample (Supplementary Fig. 2 and Source Data Table 2).

On data availability, the authors provide very helpful raw data and the code used. I would suggest to also include annotated matrices with the identified sub types in order to help to the reproducibility and the use in other studies of their findings. Other studies included interactive visualizations or directly the cell IDs and cell types annotations.

We added to Data availability github web-site links to interactive visualizations, where individual genes/samples could be explored.